



# Development of a Forced Advection Sampling Technique (FAST) for Quantification of Methane Emissions from Orphaned Wells

Mohit L. Dubey[1], Andre Santos[2], Andrew B. Moyes[2], Ken Reichl[2], James E. Lee[3], Manvendra K. Dubey[3], Corentin LeYhuelic[4], Evan Variano[1], Emily Follansbee[3], Fotini K. Chow[1], Sébastien C. Biraud[2]

[1] Department of Environmental Engineering, University of California, Berkeley, CA, 94720, USA
[2] Lawrence Berkeley National Laboratory, Berkeley, CA, 94720, USA
[3] Los Alamos National Laboratory, Los Alamos, NM, 87545, USA
[4] Ecole Normale Supérieure Paris Saclay, Paris, France

*Correspondence to*: Mohit L. Dubey (mldubey96@berkeley.edu)

**Abstract.**

Orphaned wells, meaning wells lacking responsible owners, pose a significant and poorly understood environmental challenge due to their vast number and unknown associated emissions. We propose, develop, and test a novel method for estimating emissions from orphaned wells using a Forced Advection Sampling Technique (FAST) that can overcome many of the limitations in current methods (cost, accuracy, safety). In contrast to existing ambient Gaussian plume methods, our approach uses a fan-generated flow to create a jet between the emission source and a point methane ($CH_4$) sensor. The fan flow field is characterized using a collocated sonic anemometer to measure the 3D wind profile generated by the fan. Using time-series measurements of $CH_4$ concentration and wind, a simple estimate of the $CH_4$ emission rate of the source can be inferred. The method was calibrated using outdoor controlled release experiments and then tested on four orphaned wells in Lufkin, TX, and Osage County, OK. Our results suggest that the FAST method can provide a low-cost, portable, fast and safe alternative to existing methods with reasonable estimates of orphaned well emissions over a range of leak rates.

## 1 Introduction

### 1.1 Motivation

Orphaned oil and gas wells, meaning wells lacking responsible owners, pose a significant and poorly understood environmental challenge. In the United States (U.S.) alone, there are approximately 120,000 documented orphaned wells [Merrill et al., 2023], with an estimated 310,000 to 800,000 more undocumented wells [IOGCC, 2021]. For much of the 20th century, orphaned wells were considered a non-issue compared to active wells, as they were imagined to have low emission rates, particularly when they were reported as "plugged". However, existing estimates of total orphaned well emissions are based only on direct measurements of <0.03% of known wells [Kang et al., 2023], making them a highly undersampled and uncertain source of anthropogenic methane ($CH_4$). Furthermore, it has been shown that $CH_4$ emissions from orphaned wells are currently vastly underestimated. Based on a database of leak measurements at 598 wells across the U.S. and Canada, it was found that estimated orphaned well emissions are underestimated by 150% in Canada and 20% in the U.S., making them



the most uncertain CH$_4$ source in both countries [Williams et al., 2021]. This lack of reliable emission data has resulted in increased interest in plugging orphaned wells as an important area of research for methane emissions reduction and near term
climate change mitigation.

Alongside academia, the political sphere has shown increased interest in measuring and plugging orphaned wells. The Global Methane Pledge was signed at COP26 in 2021 by 155 countries representing over 50% of global CH$_4$ emissions who committed to 30% reductions of emissions from 2020 levels by 2030 [UNFCCC Secretariat, 2022]. The U.S. has since
begun to investigate plugging orphaned wells, with an investment of $660 million in 2023 through the Department of Interior [DOI, 2023]. From 2018-2020, the average cost of plugging a single well in the U.S. ranged from $2,400 to $227,000, with an overall three-year average of $25,634 [IOGCC, 2021]. Using these numbers directly, without adjusting for inflation or overhead costs, this funding would be sufficient for plugging around 25,000 wells, or only 20% of the documented orphaned wells and a mere 3% of the upper bound of total orphaned wells. Given the high-cost of surveying and
plugging, it will be critical to prioritize wells with larger emissions to reduce the economic burden of plugging orphaned wells.

Estimating emissions from orphaned wells is challenging due to their remote locations and typically low emission rates. Based on the aforementioned database of 598 wells across the U.S. and Canada, it has been estimated that orphaned
well emission rates range from less than 1 to 48 g/h per well, with an average of around 6 g/h [Williams et al., 2021]. However, recent measurements in New Mexico and Colorado  have also shown orphaned well emissions exceeding 1 kg/h with a mean value of  120 g/h [Follansbee et al., 2024, Riddick et al., 2024]]. Still, extremely high-emitting orphaned wells are very rare and the vast majority of wells emit below the thresholds needed to observe them using current remote sensing platforms [Sherwin et al., 2024].

There are a variety of ground-based measurement approaches that can be applied to measure emissions from orphaned wells (Table 1). These range from expensive hand-held infrared cameras (FLIR) to more time-intensive mobile (OTM) [U.S. EPA, 2014] and stationary systems (SEMTECH [SEMTECH], Chamber [Williams et al. 2023], GPM [Lushie and Stockie, 2010], Vent [Ventbusters, 2023]). Unmanned aerial vehicles (UAV) have also recently been proposed as a
means of measuring wells, and preliminary results look promising [Dooley et al., 2024]. However, due to the expensive or complex nature of most of these methods only <0.03% of orphaned wells have been sampled. To overcome this data gap, new robust and fast techniques for estimating emission rates on the order of 1-10s g/h are needed (i.e. FAST).

Previous studies have investigated existing methods for quantifying methane emissions on the order of those
relevant for studying orphaned wells [Dubey et al., 2023, Riddick et al., 2023, 2022]. Table 1 shows a list of the existing technologies that can measure methane emissions in this regime and their relative costs and sensitivities. The existing





methods that are accurate and portable enough for measuring orphaned wells have other limitations, including insensitivity (FLIR), high-cost (SEMTECH), complexity and safety (Chamber, Vent, UAV, OTM), accuracy, hardware and labor costs that are summarized in Table 1. Therefore, there is a pressing need for a cost-effective, efficient, safe and accurate method
using existing sensors to estimate methane emissions for prioritizing orphaned well plugging.

| Method | FLIR | SEMTECH | Static Chamber | Dynamic Chamber | GPM | Vent | UAV | OTM | FAST |
|---|---|---|---|---|---|---|---|---|---|
| **Hardware** | $100K | $40K | $10K | $25K | >$5K | $50K | $60K | $10K | <$35K |
| **Range g/h** | >100 | <1-30,000 | 0.1-10 | 0.1-200 | >100 | >100 | 50-1500 | >50 | 1-1000 |
| **Accuracy** | Low | High | High | High | Low | High | High | Low | High |
| **Size** | Small | Small | Large | Large | Large | Large | Large | Large | Small |
| **Labor** | Low | Low | High | High | Low | High | High | Low | Low |
| **Safety** | High | Low | Low | Low | High | Low | High | High | High |

**Table 1**: Comparative assessment of commercial (FLIR, SEMTECH, Vent) and research (Chamber, GPM, UAV, OTM)
methods used to monitor fugitive methane leaks from orphaned wells. Hardware costs, detection range, accuracy, size, labor and safety are compared for each technology. The FAST costs are currently limited by sensor costs that can be reduced significantly.

80       In this paper, we propose, develop, and test a novel method for estimating $CH_4$ emissions from orphaned wells using a Forced Advection Sampling Technique (FAST) that can overcome many of the limitations of other methods, as outlined in Table 1. In contrast to existing ambient Gaussian plume methods (GPM), our approach uses a fan-generated flow to create a jet between the emission source and a point sensor. This eliminates the need for an estimate of atmospheric stability, which is required to use the GPM. Using a colocated anemometer to measure the 3D wind profile generated by the
fan, a simple estimate of the $CH_4$ emission rate of the source can be obtained. The method is calibrated using an outdoor controlled release experiment and blindly tested on four wells in Lufkin, TX, and Osage County, OK. We report results that suggest that the FAST method can provide a low-cost, portable, fast and safe alternative to existing methods to provide reasonable estimates of orphaned well emissions.



## 1.2 Mathematical Model

The physics underlying the FAST approach is based on a steady-state solution to the advection-diffusion equation. This solution, known as the Gaussian plume equation [Veigele and Head, 1978] has been widely used in the literature to perform emission inversions. However, previous studies using the Gaussian plume equation consider larger emission sources and length scales (on the order of a kilometer) than those of interest in this study [Snoun et al., 2023]. As a result, traditional GPM studies are typically dependent on parametrizations (i.e. Pasquill stability class), which are too coarse for the length

and time scales used when studying orphaned wells at smaller (on the order of a meter) length scales. Furthermore, most previous studies using the GPM approach use ambient winds as opposed to a fan-generated plume within an ambient background. In one exception to this, an approach to localize emissions using a fan-generated flow was devised by [Sanchez-Sosa et al., 2018]. However this approach was only tested indoors and did not estimate emissions for their source of interest (ethanol).


Here we outline the underlying physics of scalar transport within a jet of fan-generated turbulent flow and derive a linear equation that can be used to estimate the emission rate of a source from time-averaged centerline measurements of concentration and wind velocity within that flow.

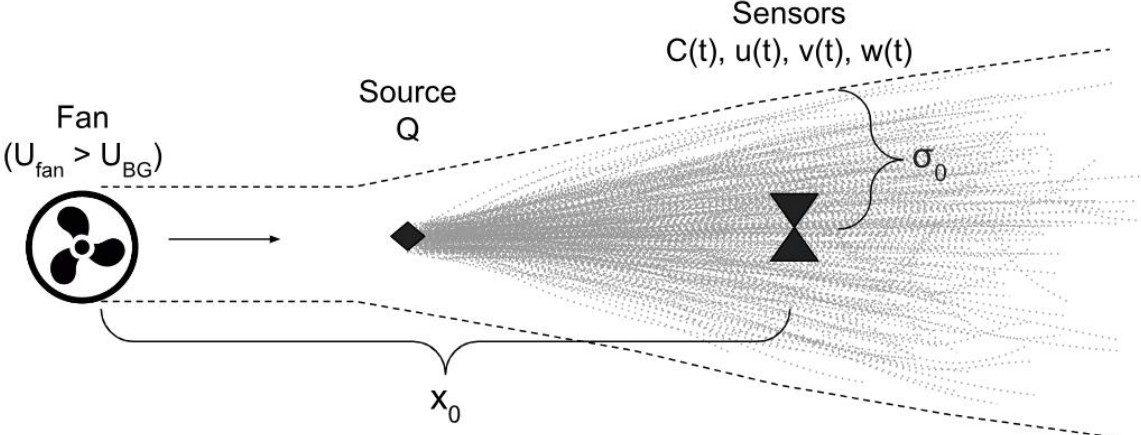

**Figure 1:** Schematic of the FAST method where an upwind fan (with mean downwind speed $U_{fan}$ larger than the background $U_{BG}$) generates a turbulent jet to advect a non-reactive gas ($CH_4$) leaking at volumetric flow rate ($Q$) from a source to downwind sensors (anemometer measuring $u(t)$, $v(t)$, $w(t)$ and $CH_4$ analyzer measuring $C(t)$).

The method assumes a constant emission source with emission rate $Q$ (g/s) positioned downstream from a fan aligned with the mean background wind direction, where the velocity of the fan flow ($U_{fan}$) is larger than that of the background wind ($U_{BG}$). Adding this fan creates an environment which is assumed to have homogeneous turbulence between





the source and the sensors. The sensors are positioned downstream along the centerline from the fan by a distance ($x_0$) and measure time series of concentration (C) and velocity (u, v, w).


The estimated emission rate ($\hat{Q}$) is calculated by integrating the scalar flux (C•u) over a circular cross-section (dA) at some downstream location.

$$Q = \oint C\,u\,dA \approx \underline{C_{CL}}\;\underline{u_{CL}}\;\pi\,\sigma_0{}^2$$
$$\hat{Q} = \underline{C_{CL}}\;\underline{u_{CL}}\;\pi\,\sigma_0{}^2 \tag{1}$$


where spatial averages of concentration and velocity are approximated with time averages (underline) of centerline measurements (subscript "CL"). This gives a radial distance ($\sigma_0$) which is approximately the effective width of the plume at the downwind distance $x_0$. This radial distance $\sigma_0$ is estimated based on a previous study of fan-generated flows [Halloran et. al, 2014] as a form of turbulent transport [Taylor, 1922]. Halloran et al. showed that the expansion of a fan-generated plume

close to the source is proportional to the square root of the downwind distance and dependent on the turbulence intensity ($i_{fan}$) and characteristic length scale ($l_{fan}$) of the fan:

$$\sigma \sim (i_{fan}\,l_{fan}\,x)^{\frac{1}{2}}$$
$$\sigma_0 = (\beta\,i_{fan}\,l_{fan}\,x_0)^{\frac{1}{2}} \tag{2}$$


Evaluating Equation 2 at location $x_0$ and combining with Equation 1, $\hat{Q}$ can be rewritten as a linear function of time-averaged centerline concentration and velocity measurements:

$$\hat{Q} = \pi\,\beta\,i_{fan}\,l_{fan}\,x_0\;\underline{C_{CL}}\;\underline{u_{CL}} = K_{FAST}\;\underline{C_{CL}}\;\underline{u_{CL}} \tag{3}$$


where the proportionality constant ($K_{FAST}$) is only dependent on constants related to the fan and the geometry of the system $\beta$ (which is treated in more detail in Appendix A):

$$K_{FAST} = \pi\,\sigma_0{}^2 = \pi\,\beta\,i_{fan}\,l_{fan}\,x_0 \tag{4}$$






## 2 Methods

### 2.1 Fan Characterization Experiments

To characterize the effectiveness of using a fan to generate a turbulent jet for the FAST method, experiments were conducted at Lawrence Berkeley National Lab (LBNL) on the afternoon of 4/16/2024 and in the morning of 5/21/2024
(Figure 2). For both experiments, a Gill Windmaster 3-D sonic anemometer was used to collect 3-D wind speed measurements at 10 Hz downwind of a Minneapolis Duct Blaster (MDB) fan with no attachments. Both the fan and the anemometer were mounted on tripods at a height of 1 meter.

During the first experiment, measurements were taken for nine-minute intervals at downwind distances of 0.5 - 5
meters for two different fan speed settings, referred to as "Low" (~3 m/s on average at a distance of 1 m) and "High" (~5 m/s on average at a distance of 1 m). The system was set up to be aligned to the background wind of ~3 m/s from West-Northwest. Despite attempts to align the fan with the dominant background wind direction, there were still persistent crosswind gusts on the order of ~1 m/s which varied as the experiment progressed.



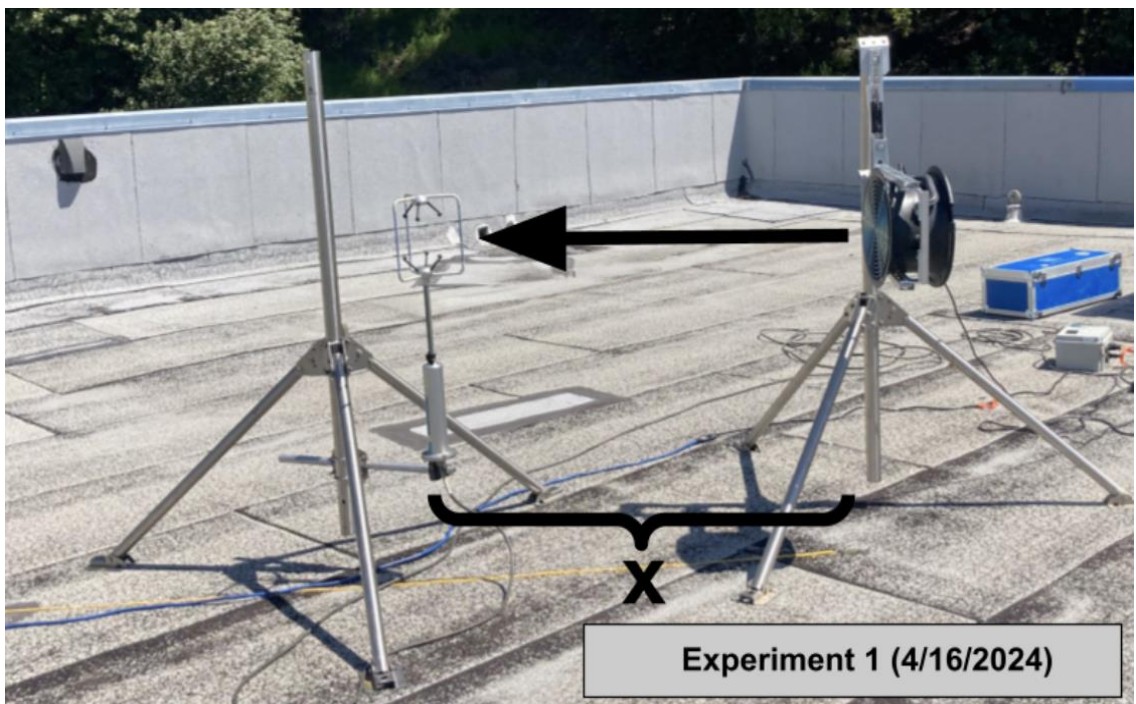

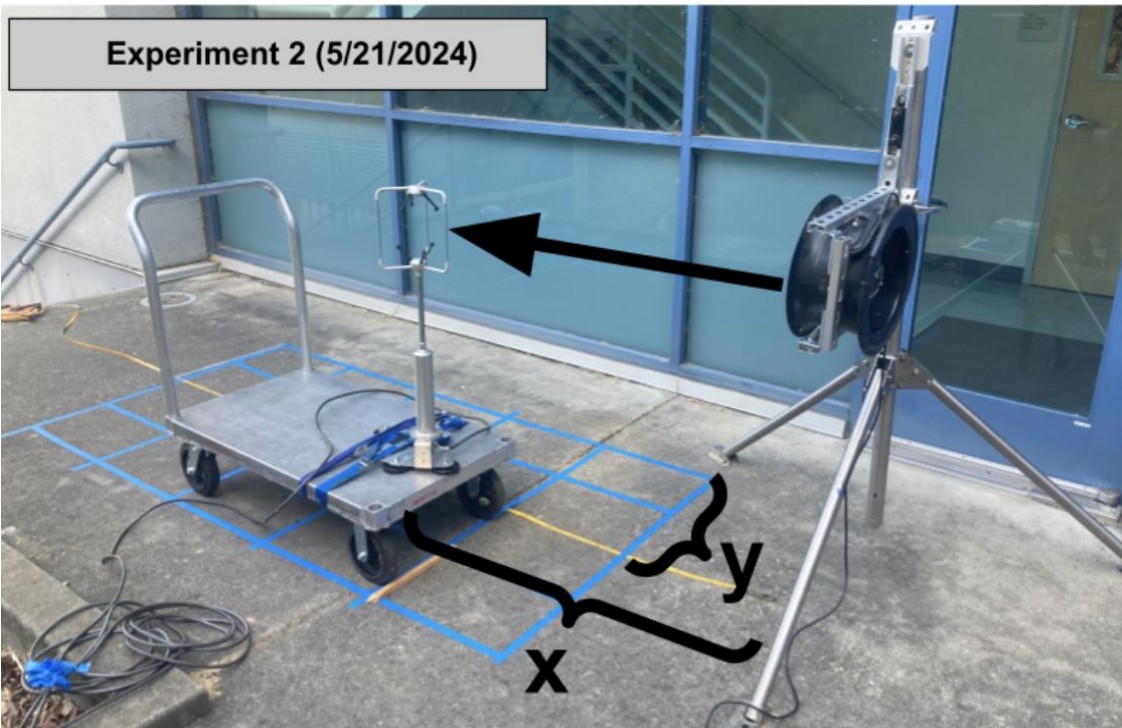


**Figure 2**: Experimental setup for fan characterization experiments, using an anemometer placed at a downwind distance x and crosswind distance y (second experiment)





**Figure 3**: a) Wind speed, b) standard deviation of wind direction, and c) turbulence intensity as a function of downwind distance x for Experiment 1 using the "Low" fan speed setting. These data have been filtered to remove any wind velocity conditions from the negative x direction







**Figure 4**: a) Wind speed, b) standard deviation of wind direction, and c) turbulence intensity of Experiment 1 using the

"High" fan speed setting. Data are filtered to remove any points coming from the negative x direction (180 degrees)





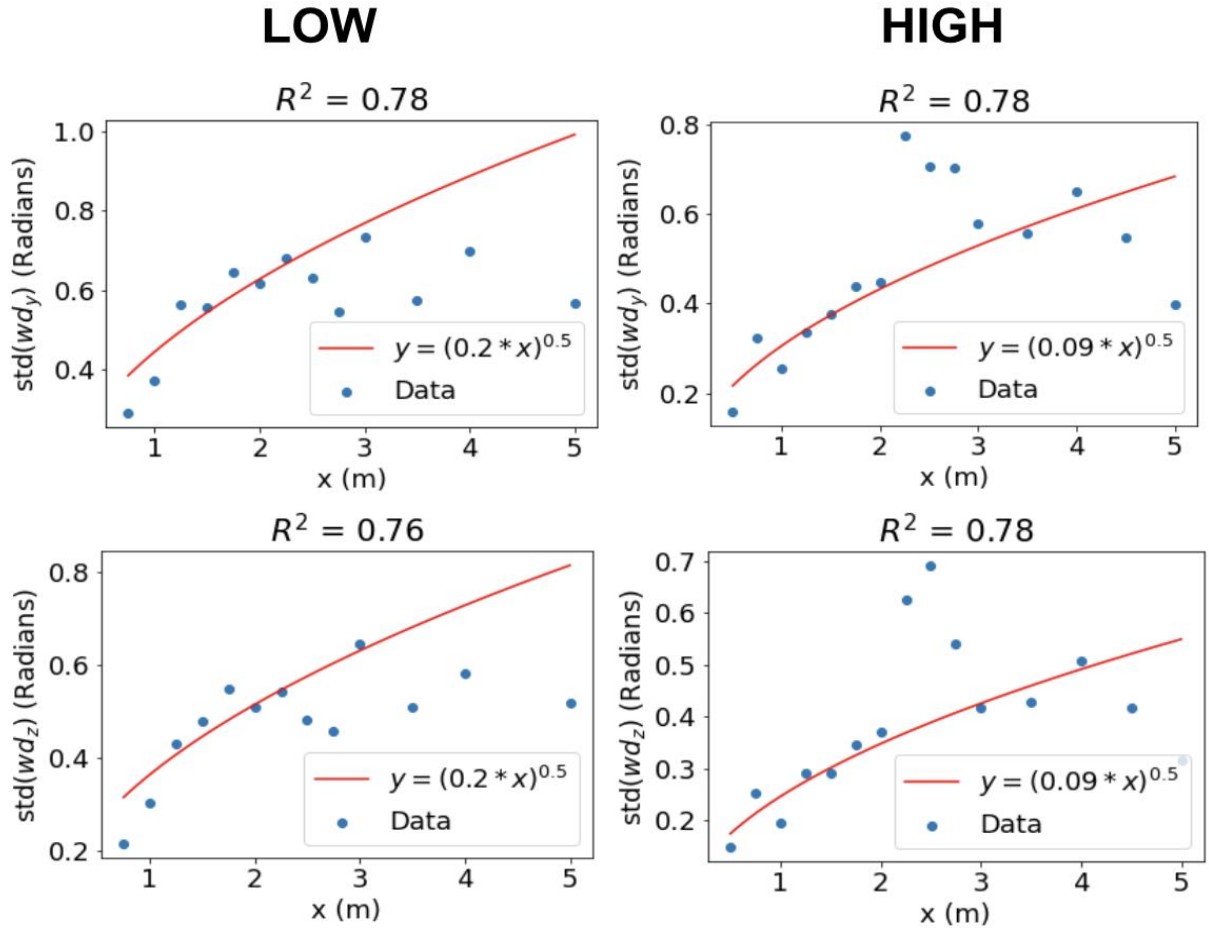

**Figure 5**: Square root fits to the standard deviations (std) of wind direction with filter angle = 180 degrees as expected by equation 2. The fits are valid in the range of 1-2 meters and depart from the square-root curve at larger downwind distances.

The results of the first fan characterization experiment are summarized in Figures 3-5. Figures 3 and 4 show the mean wind speed in 3D (u, v, w), standard deviation of wind direction in the x-y ($std(wd_y)$) and x-z ($std(wd_z)$) planes and turbulence intensity in the y ($i_y$) and z ($i_z$) directions at the range of downwind distances (x) for the "Low" and "High" fan speeds respectively. For both fan speeds, the u component of the flow starts higher than the background and decreases nearly linearly with distance until it is on the same order of magnitude as the background crosswind gust intensity (blue dashed
line). The $std(wd_y)$ and $std(wd_z)$ values are calculated using the Yamartino method [Turner, 1986] and act as an estimate of the effective width of the plume (in radians). According to Equation 2, these should both therefore grow proportional to $x^{0.5}$, which is verified for x < 2 meters in Figure 5. The turbulence intensities ($i_y$) and ($i_z$) are calculated using the means of the magnitudes of u, v and w as mean(v)/mean(u) and mean(w)/mean(u) respectively. These should be relatively constant for the fan generated flow ($i_{fan}$) and about equal if the turbulence of the flow is uniform. The uniform flow near the fan is much



more evident in the "High" fan data (Figure 4), whereas the "Low" fan measurements (Figure 3) indicate the effects of crosswind turbulence, resulting in much larger values of $i_y$ compared to $i_z$. It is also important to note that during the measurement period for the "High" fan speed, the crosswinds far exceeded their background value, resulting in a total disruption of the plume in this region (2 m < x < 3 m). The crosswinds died down later in the experiment when the anemometer was further downwind, resulting in a more stable plume for x > 3 m. Overall, we found that the fan plume

remained stable to a wide range of crosswind conditions in the range of 1 m < x < 2 m.

**Figure 6**: a-b) Mean wind speed, c-d) standard deviation of wind direction in the x-y plane, and e-f) turbulence intensity in the y-z plane of Experiment 2 for low-fan setting (left) and high-fan setting (right).




The results of the second fan characterization experiment are summarized in Figure 6. Similar to Figure 3, the top row shows the mean wind speed in the downwind direction (u), the mean of standard deviation of wind direction in the x-y ($std(wd_y)$) and x-z ($std(wd_z)$) planes, and the mean of the turbulence intensity in the y ($i_y$) and z ($i_z$) directions at a range of downwind distances (x) and crosswind distances (y) for the "Low" and "High" fan speeds (left and right, respectively).

Unlike the first experiment, these measurements allowed for variation in both x and y, allowing us to investigate the shape of the plume. The experiment was done with very little background wind (before sunrise) and in a location shielded from crosswind on one side by a wall (Fig. 2). The measurements were taken at 10 Hz for 1 minute intervals at each of the points in the x-y grid (0.5 m intervals in x for 0.5 < x < 3.0 and 0.33 m intervals in y for -0.66 < y <0.66 ) as depicted in Figure 6.

From these measurements, the MDB fan was able to generate a jet of pseudo-homogeneous turbulence at a range of downwind distances between 1 and 2 meters. Beyond 2 meters, the plume becomes unstable and can be broken easily by crosswinds, even at a "High" fan setting. Furthermore, the heat maps in Figure 6 also point to the importance of measuring along the centerline (y = 0), as the effects of crosswind turbulence increase by a large amount even when only slightly off of the centerline (y > 0.3 m). Based on these results, the controlled release experiment was conducted with the sensors at a 205 distance of 2 meters from the fan, and all of the field measurements were performed with a distance of less than 2 meters.

## 2.2 Controlled Release Experiment

To verify and estimate the relevant parameters used in the FAST method, a controlled release experiment was conducted using a range of constant methane leak rates and the SEMTECH HI-FLOW backpack system for verification. The 210 SEMTECH Hi-Flow 2 is a methane emission quantification system composed of a backpack-mounted gas analyzer and a long sampling inlet tube with a fan to sample the methane emitted by a point source. This system reports the flow of methane emitted in liters per minute (LPM)  in a range of 0.02 LPM to 730 LPM (1 g/hour - 29 kg/hour) (0.001 CFM to ~25 CFM), with an accuracy of ~10% [SEMTECH].

The measurement principle of the SEMTECH HI-FLOW relies on simultaneous measurements of air flow and methane concentration. If $F_{air}$ is the volumetric flow rate of air captured by the system (in LPM), C is the concentration of methane in ppm, $C_{background}$  is the concentration of methane of the background, and $K(T,P,\eta)$ an adjustment parameter varying with temperature (T), pressure (P), air viscosity($\eta$),  then we can express the volumetric flow rate of methane $F_{CH4}$ as:


$$F_{CH4} = F_{air} \bullet (C - C_{background} ) \bullet K(T,P,\eta) \qquad\qquad (5)$$





The velocity of the air is measured using a pitot tube, and the concentration of methane is measured using a gas analyzer
(near-IR laser absorption $CH_4$ sensor sensitive on a range of 10 ppm to 100% of $CH_4$) located in the backpack. All other
parameters such as temperature and pressure are also measured by the SEMTECH directly in the pitot tube. This system is
designed to be user friendly, as the flow measured is directly shown on the system monitor. Data are logged every second
(1Hz). For example as we can see on Figure 7, the SEMTECH measures the $CH_4$ emission rate, at a rate of one point per
second, and returns a value in liters per minute (LPM) which we converted to g/hour for more convenient use.


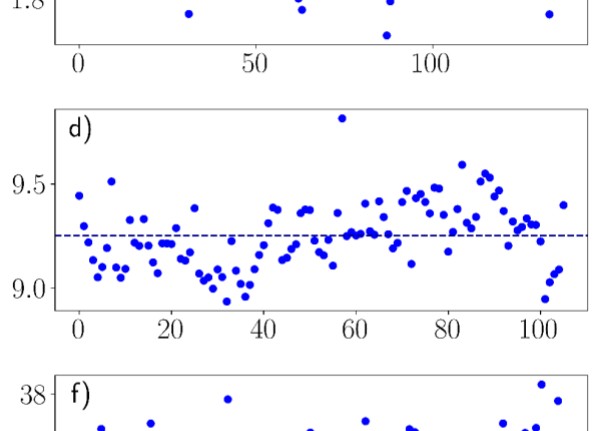

**Figure 7**: Left to right and top to bottom, shows the increasing steps measurements of controlled release emission rates from
the SEMTECH. As the control release flow rate increases, the accuracy and precision of the SEMTECH decrease.


**2.2.1 Experimental Setup**



The fan position, source position, and sampling position were all 1 meter above ground level, and 1 meter separated from each other along an axis parallel to the ground, with the source placed in between the fan and the sample point (Figure 8). A methane source was prepared by mixing 75 psi of high purity $CH_4$ with 1425 psi ultra-high purity $N_2$ in a 30 L aluminum cylinder, to obtain a blend of $5.0 \pm 0.17\%$. The source was released from the cylinder at controlled rates using a regulator plumbed through a mass flow controller programmed with set points corresponding to planned $CH_4$ emission rates of 1, 2, 5, 10, 20, and 40 g/hr. The sampling for the FAST method was conducted using a Picarro G4302 analyzer for measuring the $CH_4$ concentration and a Gill Windmaster 3-D sonic anemometer placed as physically close together as possible at the sample position. The forced advection was done using an MDB fan with no attachments. The fan, anemometer, and data collection systems were powered using a 300 Amp-hour 12V DC battery and inverter, while the Picarro analyzer ran on its internal battery.

Sensor signals from wind, ambient air temperature, pressure, and source output flow were collected using data loggers, with all data collection system clocks synchronized to within one second of UTC. The experiment began at 18:30 UTC with setup and preparation. At 20:19, the initial experiment for background ($1.99 \pm 0.36$ ppm) measurements started with no source emission. The experiment involved different flow rates with corresponding durations. For each flow rate interval, the SEMTECH measurements were conducted in 2 minutes with no fan, followed by the FAST data collection with 10 minutes without the fan, 10 minutes with the fan at low intensity, and 5 minutes of the fan at high intensity, with 5 to 10 minutes of adjustment between flow rate steps to avoid transient periods. The experiment concluded at 00:21 UTC (the following day).

| Measured Quantity | Sensor | Measurement Frequency | Data Collection System | Associated System |
|---|---|---|---|---|
| $CH_4$ (ppb) | Picarro G4302 | 2 Hz | Integrated computer running Windows 7 | FAST |
| u, v, w (m/s) | Gill Windmaster 1210-PK-085 | 10 Hz | Campbell CR1000X Datalogger | |
| Air Temperature (℃) | RM Young 41382VC | 1 Hz | | Controlled Release |
| Air Pressure (kPa) | Setra 278 | 1 Hz | | |
| Source Flow (L/min.) | Brooks Instrument GF40 | 1 Hz | Campbell CR6 Datalogger | |

**Table 2**: Equipment used during the controlled release experiment at Richmond Field Station





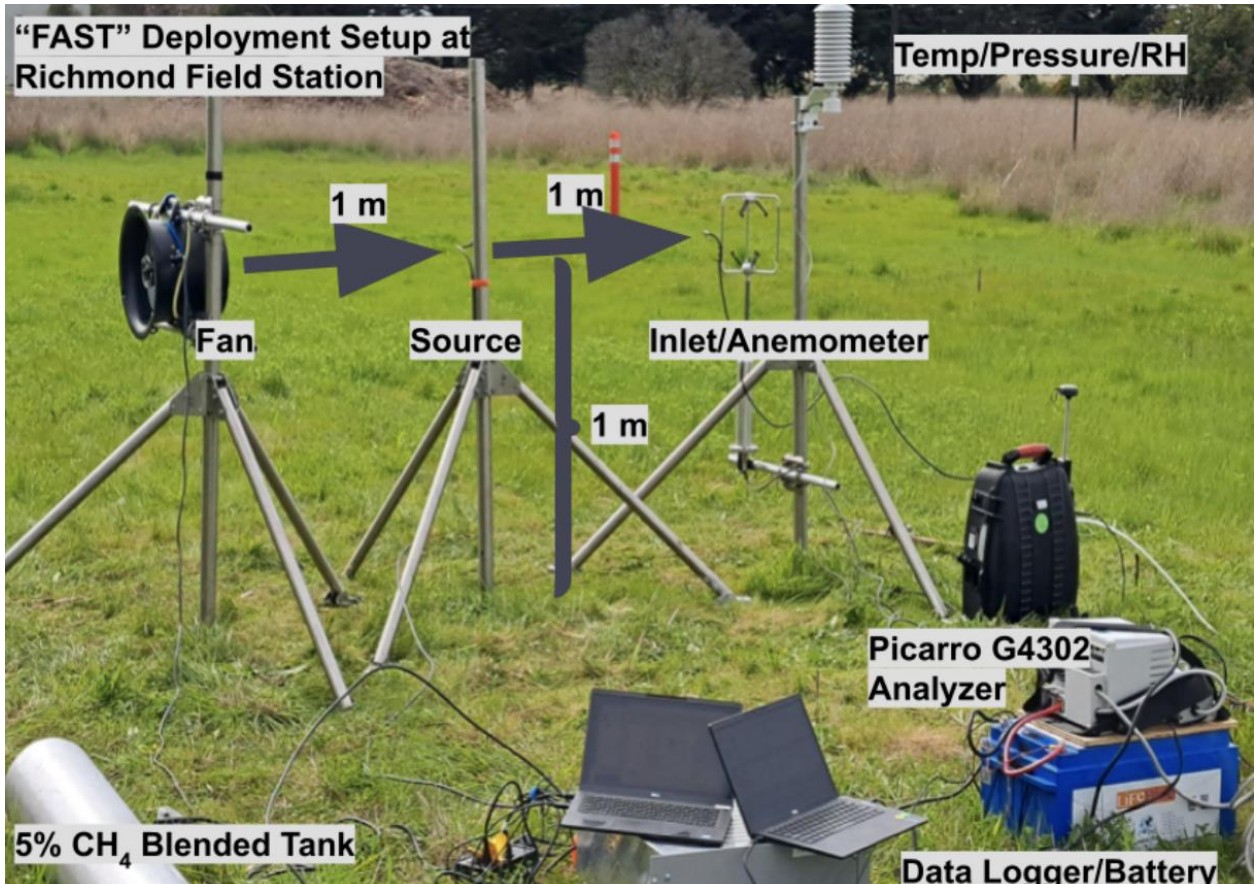

**Figure 8**: Experimental setup for the controlled release experiment at Richmond Field Station in Richmond, CA. The MDB fan, the inlet for the Picarro G4302 analyzer and the Gill Windmaster 3-D sonic anemometer were mounted at 1 m height at a distance of 2 m from one another (upper limit for the FAST method). A source of 5% methane blended with pure nitrogen was also mounted on a second tripod at a downwind distance of 1 m from the fan and outfitted with a piece of foam to ensure diffuse emissions.

**2.2.2 Stoichiometry**

The methane source leak rates in g/hr are calculated using measured quantities of source flow, ambient air temperature, pressure, and assumed constant source concentration. The measured quantities reported for each step were averaged over each measurement. The source leak rate Q is described in terms of measured quantities and known constants.

$$Q = C\,\rho\,\kappa \qquad\qquad (6)$$

where, $C$ is the $CH_4$ concentration from the source tank at $0.05 \pm 0.0017$ [mol $CH_4$ / mol air], $\rho$ is the $CH_4$ mass density [g/L] at measured ambient temperature and pressure, and



275          $\kappa$ is the corrected output mass flow of the source gas.

The CH$_4$ mass density is calculated in terms of measured qualities of ambient air pressure $P$ and temperature $T$ as

$$\rho = (M_{CH4}/R)(P/T)$$

where, $M_{CH4}$ is molar mass 0.01604 [Kg / mol] of CH4, and $R$ is the universal gas constant 8.31446 [(L · kPa)/(K ·

mol)]. The corrected output mass flow $\kappa$ is calculated from the measured flow rate $\kappa_{std}$ reported at standard

temperature $T_{std}$ of 293 Kelvin and measured ambient temperature $T$ as

$$\kappa = \kappa_{std}(T/T_{std})$$

Rewriting Q in terms of measured quantities we find:

285                    $$Q = \alpha C P \kappa_{std} \tag{7}$$

where,

$\alpha = [M_{CH4}/(RT_{std})]$

The source leak rate uncertainty $\sigma_Q$ (shown as error bars on Q estimates) is estimated from uncertainties in source

concentration $C$, measured quantities of ambient air pressure $P$ and output flow $\kappa_{std}$

$$\sigma_Q = \alpha\sqrt{(P\kappa_{std}\sigma_C)^2 + (C\kappa\sigma_P)^2 + (CP\sigma_\kappa)^2} \tag{8}$$

where the uncertainties $\sigma_P$ and $\sigma_\kappa$ are standard deviations of averaged data from the measurement windows. The time series
of flow rates and measured atmospheric pressure during the course of the experiment are shown in Figure 9.

Similarly, the uncertainty in the flow rate estimated by FAST method can be written as:

$$\sigma_{\hat{Q}} = \sqrt{(K_{FAST}\,\underline{C_{CL}}\,\sigma_{u_{CL}})^2 + (K_{FAST}\,\underline{u_{CL}}\,\sigma_{c_{CL}})^2 + (\underline{C_{CL}}\,\underline{u_{CL}}\sigma_{K_{FAST}})^2} \tag{9}$$

where $\sigma_{u_{CL}}$ is the standard deviation of the wind speed in the downwind direction, $\sigma_{c_{CL}}$ is the standard deviation of the

concentration measurements and $\sigma_{K_{FAST}}$ is the standard error in the estimate of K$_{FAST}$ as shown in Figure 11.



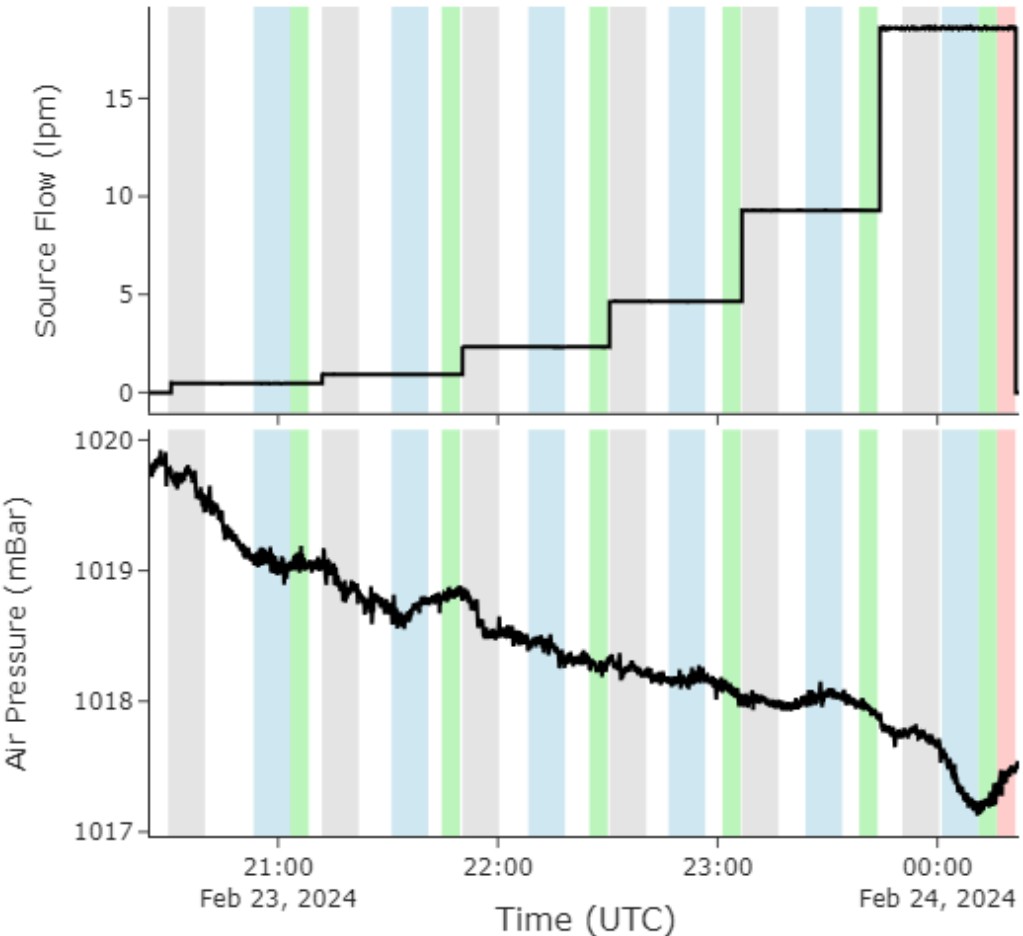

**Figure 9**: Time series of the output flow rate (liters per minute) throughout the Richmond controlled release experiment with shaded areas indicating the set measurement windows: gray for no fan state, blue for fan on at low speed, green for fan on at high speed, and red for maximum fan speed at the end of the experiment.

### 2.2.3 Experimental Determination of $K_{FAST}$

Results from the Richmond Field Station controlled release experiment are summarized in Table 3 and Figure 10. Using equations 7 and 8, an estimate of the actual source rate (Q) was obtained which nearly matched the intended target rates of 1, 2, 5, 10, 20 and 40 g/hr. The SEMTECH HI-FLOW performed very well during the controlled release study, almost matching the exact values derived from stoichiometry. The FAST method estimates (generated using 10 minute averages) also match the source rate quite well, however with much larger uncertainties than the SEMTECH. Without the fan (No Fan), the FAST method tends to overestimate the lower range (1-5 g/hr) and severely underestimate the upper range





(5-40 g/hr). This is greatly improved via the use of the fan, with the Low Fan setting performing better in the upper range and the High Fan setting performing better in the lower range. These discrepancies could also be due to fluctuations in the background wind throughout the experiment which may have biased the results.


| Source Rate $Q \pm \sigma_Q$ (g/hr CH$_4$) | SEMTECH (g/hr CH$_4$) | FAST (No Fan) (g/hr CH$_4$) | FAST (Low Fan) (g/hr CH$_4$) | FAST (High Fan) (g/hr CH$_4$) |
|---|---|---|---|---|
| $0.93 \pm 0.03$ | $0.96 \pm 0.03$ | $1.59 \pm 4.41$ | $0.47 \pm 0.90$ | $0.91 \pm 1.34$ |
| $1.86 \pm 0.06$ | $1.89 \pm 0.05$ | $0.72 \pm 3.34$ | $1.57 \pm 2.61$ | $2.78 \pm 3.68$ |
| $4.66 \pm 0.17$ | $4.62 \pm 0.08$ | $6.23 \pm 19.26$ | $3.23 \pm 5.43$ | $3.88 \pm 5.87$ |
| $9.33 \pm 0.32$ | $9.25 \pm 0.16$ | $0.74 \pm 7.34$ | $8.55 \pm 13.57$ | $8.80 \pm 12.62$ |
| $18.67 \pm 0.63$ | $18.30 \pm 0.33$ | $1.38 \pm 11.21$ | $13.2 \pm 21.62$ | $22.25 \pm 29.89$ |
| $37.32 \pm 1.27$ | $36.90 \pm 0.53$ | $25.96 \pm 94.71$ | $31.86 \pm 52.12$ | $28.45 \pm 44.65$ |



**Table 3**: Comparison of the desired (target) and true (stoichiometric) release rate with those measured by the SEMTECH and FAST method (with 180 degrees of filtering) during the controlled release experiment at Richmond Field Station.




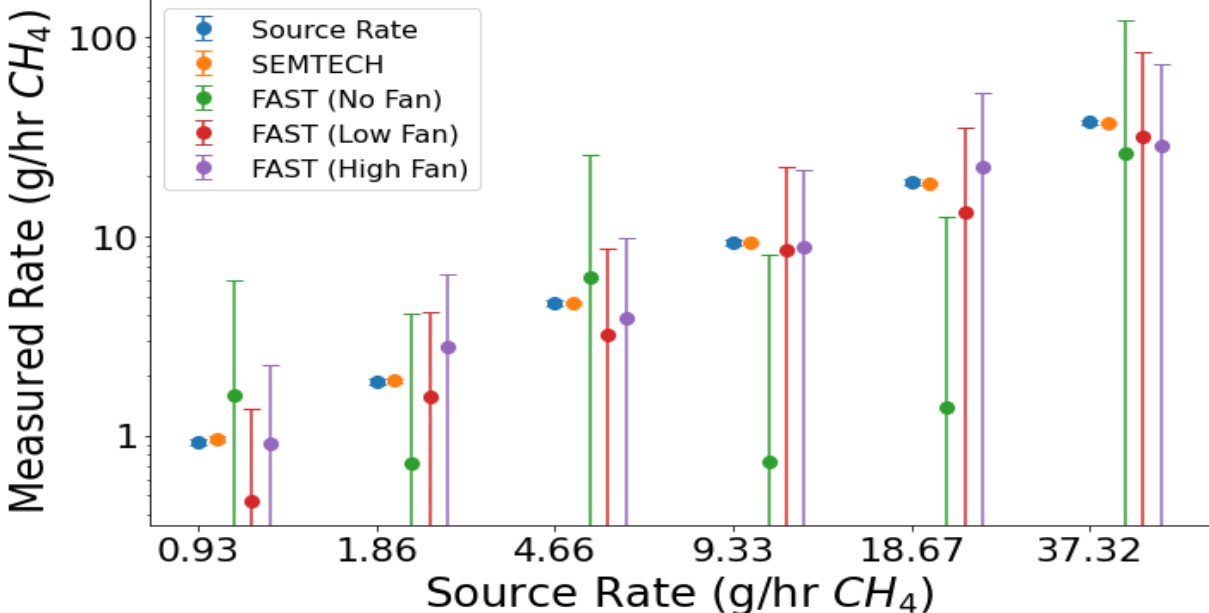

**Figure 10:** Comparison of the SEMTECH (orange) and FAST method (No Fan in red, Low Fan in green, High Fan in purple all using 180 degrees of filtering) during the controlled release experiment at Richmond Field Station with the true (stoichiometric) release rate (source rate, blue).





**Figure 11**: Measured mean C * U (along x) vs. known Q (along y) for the control release experiment used to determine the values of K$_{FAST}$ (slopes) for a range of fan speeds and filtering angles.

By using the known values of Q from stoichiometry (source rate) and the measured values of C and u during the controlled release experiment, the values of K$_{FAST}$ are estimated as defined in Equation 3. By inverting Equation 3 to solve for K$_{FAST}$ , $K_{FAST} = \frac{C_{CL}\ u_{CL}}{Q}$ where the known value of Q and 10 minute averages of C$_{CL}$ and u$_{CL}$ are used to estimate K$_{FAST}$. The resulting values for K$_{FAST}$ are shown as the slopes of the lines in Figure 11 along with the uncertainties resulting from standard error estimates on the linear regression used to generate the line of best fit. As is to be expected, the No Fan





scenario has a much higher value of $K_{FAST}$ with higher overall uncertainty due to the variation of the natural wind direction and speed. Without filtering the data by wind direction, the $K_{FAST}$ values are larger (likely due to more dispersion from crosswinds). Furthermore, $K_{FAST}$ values at the low and high fan speeds do not agree, although $K_{FAST}$ is theoretically independent of fan speed (per Equation 4). As more and more crosswind is filtered (Filter Angle approaches 360 degrees), the low and high fan speeds converge to a $K_{FAST}$ of around 0.2, implying that the plume width from the fan at a downwind distance of 2 meters is around 0.28 meters, which is slightly larger than the diameter of the fan (0.25 m), as expected. All fits are done with a 0 intercept and standard errors are used to estimate the uncertainty of $K_{FAST}$. By filtering out all crosswind interference (Filter Angle = 300), Equation 4 holds and the value of $K_{FAST}$ for the system converges to ~0.2 m$^2$. Using Equation 4 and the known values for $\beta = 1, i_{fan} = 0.17, l_{fan} = 0.13\ m, x_0 = 2\ m$, the effective width of the fan-generated plume for the controlled release experiment is estimated to be $\sigma_0 = \sqrt{\beta\ i_{fan}\ l_{fan}\ x_0} = 0.21\ m$.

## 2.3 Field Experiments

The FAST method was tested on four wells during two field campaigns, two in Texas (6 and 7 February) and two in Oklahoma (14 March), during the spring of 2024. For all field measurements, a similar setup was used to that in the Richmond Field Station controlled release experiment. During the experiments in Texas, the background wind velocity was < 1 m/s, so only the Low Fan setting was used for the two wells. In Oklahoma, background wind speeds were much higher than those in Richmond, so both the Low Fan and High Fan settings were used. Figure 12 shows images of the four wells discussed in the paper with the FAST method setup. For each well, SEMTECH and FAST measurements were taken; FLIR measurements were taken in Texas only.



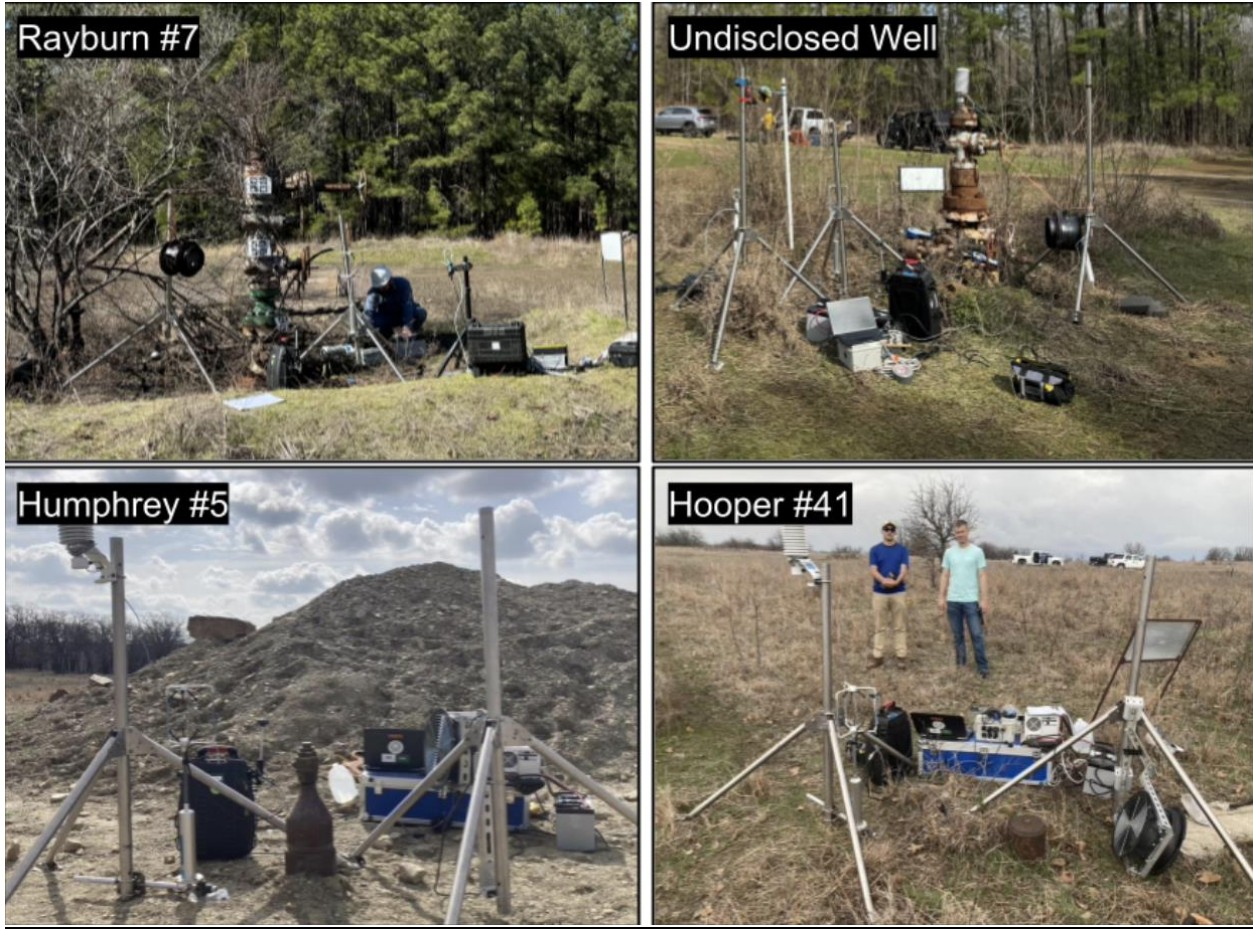

**Figure 12:** Wells measured in Lufkin, Texas (top) and Barnsdall, Oklahoma (bottom) using the FAST method. Both wells in Texas were of the "Christmas tree" variety (multiple potential leak points) and were measured using No Fan and Low Fan speeds because the background wind speeds were < 1 m/s. Both wells in Oklahoma were lower to the ground, had only one leak point and were measured with No Fan, Low Fan and High Fan speeds due to the higher background winds (> 1 m/s).

### 2.3.1 Texas Field Campaign

The first field campaign that measured orphaned wells using the FAST method took place in February 2024 in collaboration with multiple agencies. The U.S. Forest Service (USFS) invited the U.S. Department of Energy's Consortium Advancing Technology for Assessment of Lost Oil and Gas Wells (CATALOG) team to help measure and assess emissions from certain wells being plugged using funds from the Bipartisan Infrastructure Law (BIL). This effort involved collaboration between USFS, DOE, and the Texas Railroad Commission to make the most of federal and state well-sealing activities. The USFS was mainly focused on measuring methane emissions at documented abandoned wells in the Texas



National Forests and Grasslands, specifically in the Angelina and Sabine Districts. The joint field work and emission results were significant for meeting the reporting requirements of the BIL. The FAST method was deployed in the field campaign to understand emission patterns better and help allocate sealing funds more efficiently.


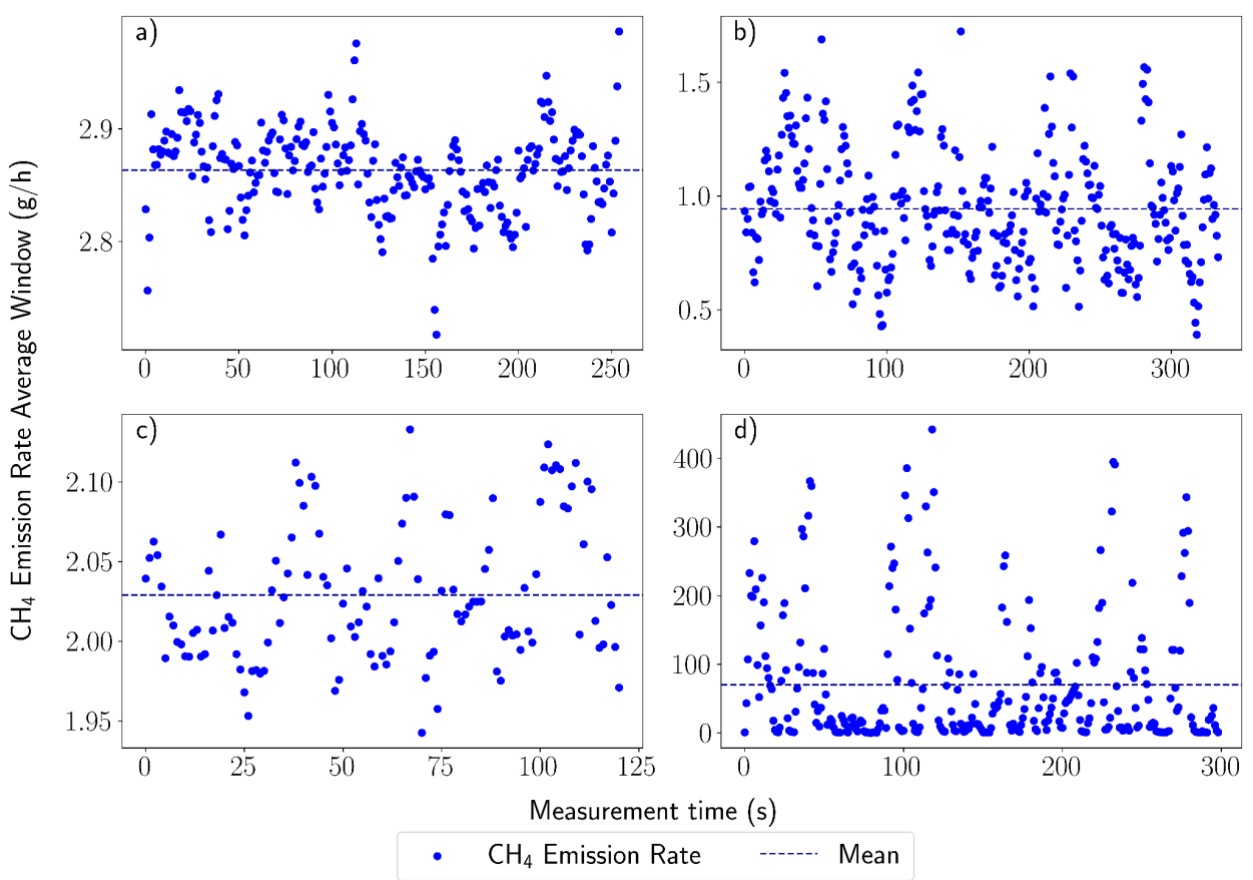

**Figure 13:** SEMTECH HI-FLOW measurements (time series and mean) from the four orphaned wells used in the FAST
method validation.

**2.3.1.1 Rayburn #7**

Rayburn #7 is an oil production well identified by API number 4200530245 and associated with district/lease
number 06/13688. Its geographical coordinates are 31.0865, -94.1974, and its total depth is 12,927 feet. On February 6, 2024 during the initial detection of Rayburn #7, a small leak was found from a threaded port on a valve junction 1.2 meters above





the ground. The well is situated in a large clearing with a gravel pad and other infrastructure, surrounded by an embankment. A plastic spill tub and several 500-gallon drums were observed close to the wellhead. Furthermore, a compressor station and separation/storage infrastructure are located in the corner of the clearing. No leak was detected on any of the other

infrastructure in the area. A Gill R3-50 sonic anemometer was placed at the height of 1.2 meters and 0.94 meters away from the methane source, alongside with the inlet to the Picarro G4302 methane detector. Sampling commenced at 12:20 under ambient conditions for 60 minutes. The background methane concentration was 2.11 ppm. Following this, there were two more sets of sampling periods: 30 minutes each for ambient conditions and low fan conditions. The SEMTECH measured an emission rate of 2.86 grams per hour, with a standard deviation during the averaging period of ± 0.04g/h (Figure 13a). The

FLIR camera could not provide a clear visual indication of the leak.

### 2.3.1.2 Undisclosed Well

The methane emission detection and monitoring experiment at this undisclosed location identified two leak points on one wellhead. The FLIR did not detect any emissions from either of the leak sources. There was a small leak at the end of a main pipe flange and a more significant leak in a connection thread on the same flange, which was the primary point source. The FAST system was set up at 11:45 UTC on February 7, 2024, pointed at 315 degrees N, with the fan turned off. The fan was located 58 cm above the ground and 87 cm upwind from the source,  which is within the range of <1 m. The

Gill R3-50 sonic anemometer and Picarro G4302 gas analyzer inlets were positioned 73 and 71 cm above the ground and 97 and 95 cm horizontally from the source, respectively. The primary source was at a height of 47 cm. The background methane concentration was 2.08 ppm. The experiment with the fan turned on started at 13:22 UTC. Data was collected for two 15-minute periods with the fan on and two 15-minute periods with the fan off. Analogous to the Rayburn #7 experiment, the anemometer orientation was set in a manner that 0 degrees represents the direction where it is facing the upwind source and

fan line. For this experiment, wind direction had favorable conditions, which led to significant data acquisition for the periods without the fan. The SEMTECH measured an emission rate of 0.95 grams per hour, with a standard deviation during the averaging period of ± 0.25 g/h (Figure 13b).

### 2.3.2 Oklahoma Field Campaign

The second field campaign that measured orphaned wells using the FAST method took place in March 2024 in collaboration with multiple agencies. The US DOE CATALOG team focuses on developing technologies to detect and characterize undocumented orphan wells (UOWs), especially in Osage County, Oklahoma. The initiative in Osage County

involves collaboration with the Osage Nation to identify and measure methane emissions from these UOWs.



**2.3.2.1 Humphrey #5**

Humphrey #5 is an orphaned well that was located by our team during surveying on March 14th, 2024. The well
was located next to an active well with a local operator. The FAST method was set up and the well was measured. During
the measurement period, the operator arrived and claimed he could fix the leak coming from the well. We allowed the
operator to attempt to fix the leak, but this invalidated the FAST measurements before, as the SEMTECH was not measured
in advance. After the "fix", the well was still leaking, so the FAST method was adjusted to a closer geometry and measured
for 10 minutes at each fan speed (No, Low, High) as shown in Figure 14. The leak was from the top cap of the well head at a
height of 0.62 m, and the sensors were positioned downwind at 0.4 m from the leak and a height of 0.65 m. The fan was set
up at a height of 0.6 m and an angle of 5 degrees upward (to generate a plume that passed through the anemometer) at 0.5 m
upwind of the well. The SEMTECH measured an emission rate of 2.03 grams per hour, with a standard deviation during the
averaging period of ± 0.04 g/h (Figure 13c).

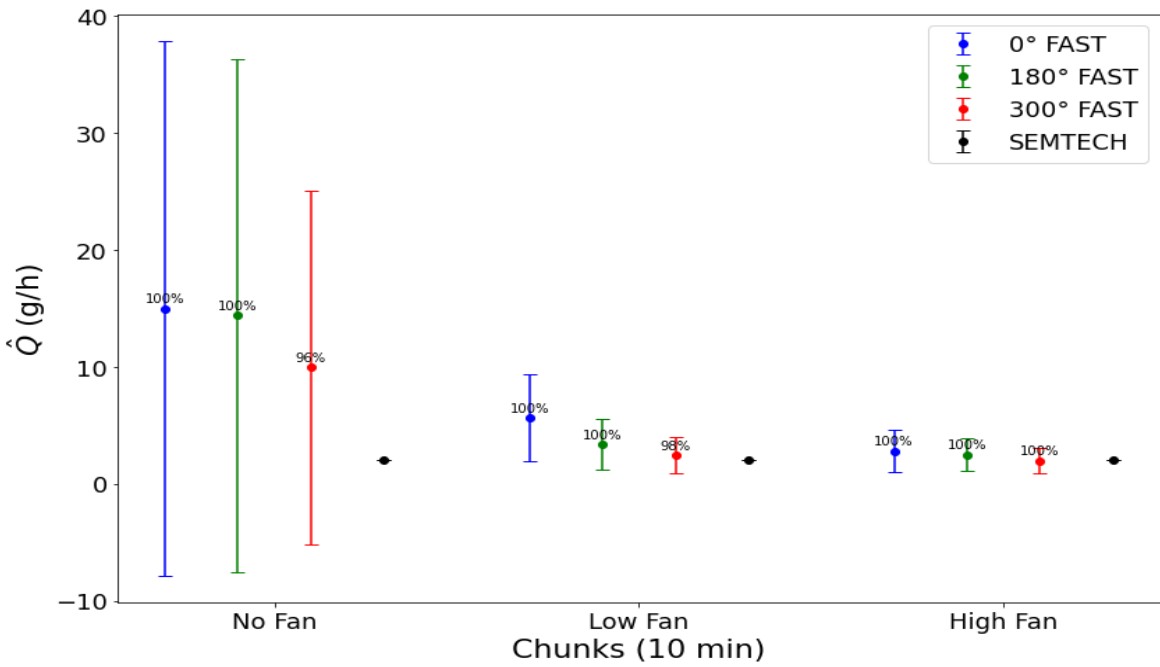


**Figure 14**: Estimated leak rates ($\hat{Q}$) and uncertainties ($\sigma_{\hat{Q}}$) from FAST method measured in 10 minute increments (colored)
and SEMTECH (black) for Humphrey #5, labeled by percentage of data kept after filtering. For the "No Fan" setting (left
most), the estimate is very uncertain and much higher than the SEMTECH.




### 2.3.2.2 Hooper #41

Hooper #41 is another UOW that was also discovered by the team on March 14th, 2024 near Barnsdall, OK. The leak was from the top cap of the well head at a height of 0.25 m, the sensors were positioned downwind at 0.88 m from the leak and a height of 0.65 m. The fan was set up at a height of 0.27 m and an angle of 24 degrees upward at 0.4 m upwind of

the well. Interestingly, this well seemed to have a variable leak rate, resulting in a very high uncertainty on the SEMTECH. The SEMTECH measured an intermittent averaged emission rate of 70.14 g/h, with a large standard deviation during the averaging period of ± 95.47 g/h (Figure 13d). The FAST method was used in 10 minute intervals for No, Low, and High fan settings. Due to the highly variable nature of the well, these measurements were repeated in the same intervals for comparison (Figure 15). Results show that the FAST method achieves a much better accuracy and estimates the well to only

emit around 10 +/- 10 g/h as opposed to 70 +/- 90, which the SEMTECH reported.

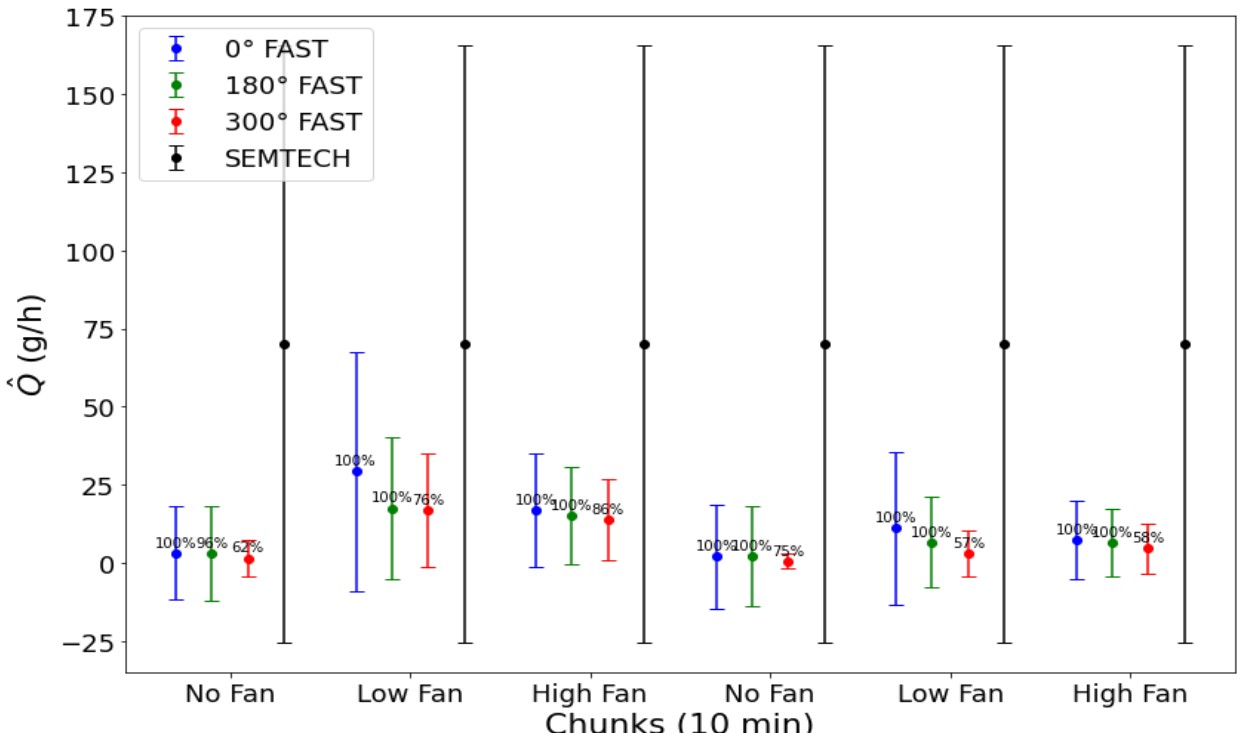

**Figure 15**: Estimated leak rates ($\hat{Q}$) and uncertainties ($\sigma_{\hat{Q}}$) from FAST method measured in 10 minute increments (colored) and SEMTECH (black) for Hooper #41, labeled by percentage of data kept after filtering. Due to the high variability of the

well, it was measured twice with the FAST method at 10 minute increments (sequentially). Here, the SEMTECH did not get a good reading due to the instability in the $CH_4$ concentration of the sampling volume.





## 3 Results and Discussion

The results of the field campaigns are summarized in Table 4 and Figure 16. For each of the four wells measured, the FAST method results are based on 10-minute averages and $K_{FAST}$ values corresponding to the fan speed and various levels of filtering (0, 180 and 300 degrees) as shown in Figure 11. The uncertainty in the FAST estimates is calculated using Equation 9. Overall, the FAST method agrees with the SEMTECH for both the Low and High fan settings but not for the No Fan settings. These uncertainties decrease with a larger filtering angle and a higher fan speed.


**Figure 16**: Estimated leak rates ($\hat{Q}$) and uncertainties ($\sigma_{\hat{Q}}$) for the four wells shown in Figure 13 from the SEMTECH (black and gray) and FAST method (colored). SEMTECH is able to get very accurate readings for all wells except Hooper #41



which was a highly variable well. The FAST method (with the fan on) works almost as well as the SEMTECH with 300
degrees of filtering and provides a more accurate reading than the SEMTECH for Hooper #41, likely due to the larger
sampling volume.

| Date | Well ID | SEMTECH (g/hr) | FAST (0 Filter) (g/hr) | FAST (180 Filter) (g/hr) | FAST (300 Filter) (g/hr) |
|---|---|---|---|---|---|
| 2024-02-06 | Rayburn #7 (Lufkin) | 2.9 ± 0.0 | Low: 5.2 ± 4.5 No: 0.8 ± 2.8 | Low: 3.3 ± 2.7 No: 0.9 ± 3.1 | Low: 2.6 ± 1.9 No: 0.6 ± 1.8 |
| 2024-02-07 | Undisclosed Well (Lufkin) | 1.0 ± 0.3 | Low: 1.0 ± 1.4 No: 0.4 ± 2.4 | Low: 0.7 ± 0.9 No: 1.1 ± 3.8 | Low: 0.5± 0.7 No: 1.3 ± 3.4 |
| 2024-03-14 | Humphrey #5 (Barnsdall) | 2.0 ± 0.04 | High: 2.8 ± 1.8 Low: 5.6 ± 3.7 No: 15.0 ± 22.8 | High: 2.5 ± 1.5 Low: 3.4 ± 2.2 No: 14.4 ± 21.9 | High: 2.0 ± 1.1 Low: 2.5 ± 1.5 No: 10.0 ± 15.1 |
| 2024-03-14 | Hooper #41 (Barnsdall) | 70.1 ± 95.5 | High: 12.1 ± 15.3 Low: 20.2 ± 31.4 No: 2.6 ± 15.8 | High: 10.8 ± 13.3 Low: 12.0 ± 18.6 No: 2.6 ± 15.6 | High: 9.3 ± 10.4 Low: 9.9 ± 12.7 No: 1.0 ± 4.0 |

**Table 4:** Estimated leak rates ($\hat{Q}$) and uncertainties ($\sigma_{\hat{Q}}$) for the four wells shown in Figure 13 from the SEMTECH and
FAST method. SEMTECH can get very accurate readings for all wells except Hooper #41 which was a highly variable well.
The FAST method (with the fan on) works almost as well as the SEMTECH with 300 degrees of filtering and provides a
more accurate reading than the SEMTECH for Hooper #41, likely due to the larger sampling volume.





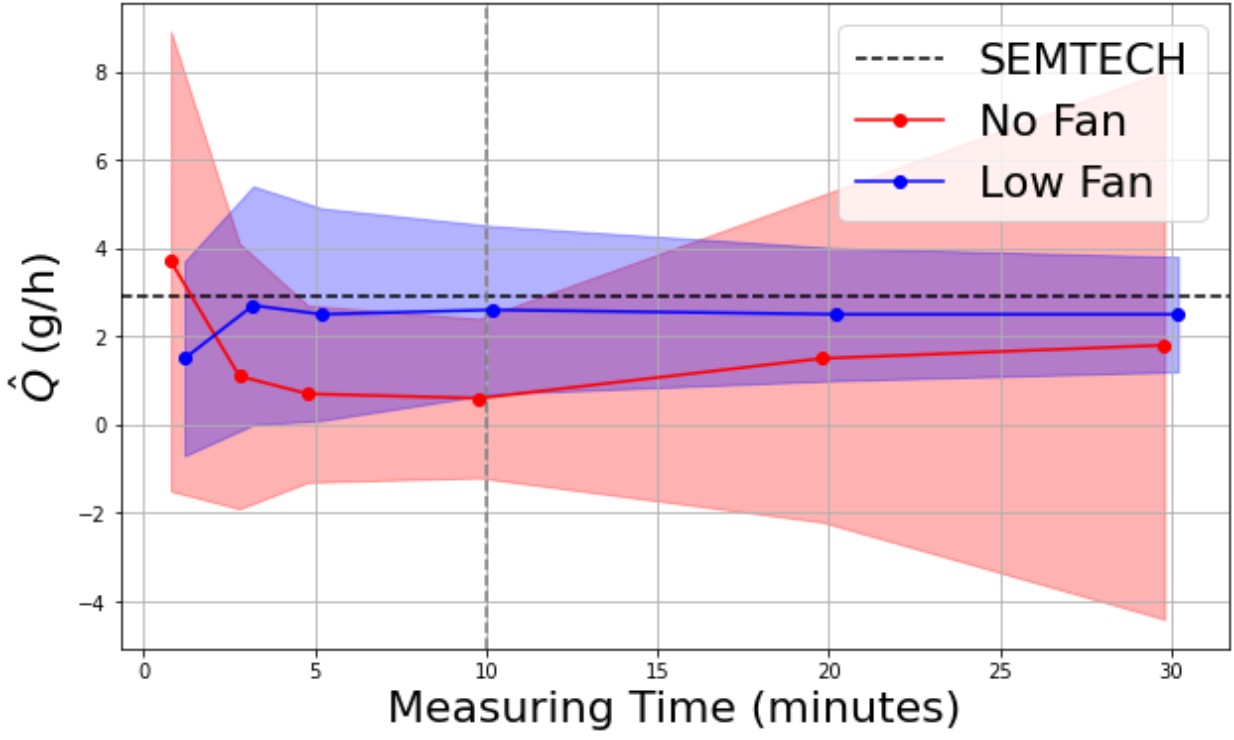

**Figure 17**: Estimated leak rates ($\hat{Q}$) and uncertainties ($\sigma_{\hat{Q}}$) from the FAST method for Rayburn #7 as a function of the sampling time used to make the estimate. Without a fan, the measurement is highly uncertain throughout, except for the range from 5-10 minutes. With the fan, the measurement accuracy gets higher with increasing measurement time, but the mean value stays roughly constant above 3 minutes. This shows that the FAST method can be done as quickly as the SEMTECH and accuracy only increases with increased measuring time.

Figure 17 shows the results of the FAST method at the Low fan (blue) and No fan (red) settings on Rayburn #7, which was measured in 30 minute intervals, as well as the SEMTECH estimate (black dashed line). For the Low fan setting, the SEMTECH value is always within the uncertainty of the FAST method, even if only the first minute of data is used. The mean rate improves when the measuring time is increased to three minutes, but the error bars remain large. For measuring times larger than three minutes, the error bars decrease nearly linearly with increased measuring time, while the mean stays relatively constant. The No Fan results, on the other hand, do not match the SEMTECH well for measuring time shorter than 10 minutes. As expected, as the measuring time increases, the mean value of the FAST method for No Fan gradually approaches the SEMTECH, but the error bars also increase over time. This shows that the FAST method, even at the Low Fan measurement, can make reliable measurements on the order of 1 - 5 minutes, which is as fast or faster than the time taken to use the SEMTECH.

The total cost of the sensors used in this study, a Picarro G4302 for concentration and Gill Windmaster 1210-PK-085 for wind, are about the same cost as a SEMTECH HI-FlOW. However, the FAST method can be done without using 3D wind measurements. By replacing the 3D anemometer with a 1D anemometer, the cost of the FAST method can be decreased with minimal loss in accuracy. Effectively, using a 1D anemometer would limit the filter angle to be up to 180 degrees, which has marginally worse accuracy than filtering by 300 degrees. Furthermore, the type of methane sensor can also be optimized to a more reasonable price point as the $CH_4$ signals near source are high (e.g. > 1 ppm for leaks > 1 g/h) and can be measured by less sensitive and more affordable near IR or non dispersive absorption or solid state methods. The parts per billion/sec sensitivity Picarro G4302 has limitations for the FAST method due to its low range (saturates above 800 ppm in methane mode). Future work will focus on investigating a wide variety of technologies (NDIR, Off Axis ICOS, etc.) to find a more cost effective and reliable methane sensor for wide scale FAST method deployment.

Besides its potential for being lower cost, the FAST method has other advantages over the existing technology (FLIR and SEMTECH). First, the FLIR camera is insensitive to small leaks and unable to detect most diffuse emissions and is unable to quantify emissions accurately [Zeng and Morris, 2019]. Furthermore, Figure 15 shows an example of a well for which the FAST method has a lower measurement uncertainty (~30%) compared to the SEMTECH. This is because the SEMTECH has a very limited sampling volume due to its closed design with a narrow tube, as opposed to the FAST method which takes advantage of the larger mixing volume of the fan-generated plume. Finally, while our existing proof-of-concept FAST hardware is currently heavier and more complex to operate than a SEMTECH, it could be replicated with a battery powered fan mounted to a tripod or a backpack vacuum blower, making it very similar to the size and labor requirements of the SEMTECH.

## 4 Summary and Conclusions

We have shown that using a fan-generated flow (forced advection) to create a jet between the emission source and a point methane ($CH_4$) sensor and measuring 3D wind profiles using a sonic anemometer and $CH_4$ concentration with a gas analyzer (sampling technique), a simple estimate of the $CH_4$ emission rate of the source can be inferred (FAST method). The results from the FAST method across various controlled release and field campaigns demonstrate its potential for rapid methane emission estimation, particularly in identifying and prioritizing orphan wells for plugging. As outlined, the FAST method consistently provided reasonable estimates of leak rates when fan speeds and filtering were applied appropriately, performing similarly to the commercially available methods (SEMTECH) and outperforming others (FLIR). Notably, the method's performance improves with increased fan speed and filtering angle. For instance, in the case of Rayburn #7 in Lufkin, Texas, the FAST method at the Low Fan speed consistently produced leak rate estimates that were within the





uncertainty bounds of the SEMTECH values after just a few minutes of measurement. Without the use of a fan, the results
showed much greater uncertainty, highlighting the importance of airflow in stabilizing methane dispersion for accurate
estimation.

In the Texas and Oklahoma field campaigns, the FAST method was able to provide accurate and rapid readings
under varying environmental conditions with errors on the order of 95% of the emission rate across a variety of wind
conditions and leak rates. In Texas, where wind speeds were low, only the Low Fan setting was used, and FAST results
aligned closely with SEMTECH, within 10%. Higher wind conditions in Oklahoma, however, required both Low and High
Fan settings to account for greater natural dispersion. The FAST method performed especially well on wells such as Rayburn
#7 and Humphrey #5, offering comparable accuracy to SEMTECH, while providing lower uncertainty in the case of the
highly variable Hooper #41, where SEMTECH struggled with the well's fluctuating leak rates from close point sources. This
advantage of the FAST method arises from its larger sampling cross-section and volume, which captures a broader
representation of methane concentration, particularly in environments with variable and multiple leaks.

The FAST method provides a cost-effective, scalable, and practical alternative to existing technologies such as
SEMTECH and FLIR for identifying high-priority orphan wells. The combination of fan-induced plume dispersion and real-
time methane measurement allows for quick assessments (on the order of minutes), making it well-suited for large-scale
monitoring. While FAST may not reach the same precision as SEMTECH under all conditions, especially without adequate
filtering, its ability to balance speed, cost, and accuracy makes it a viable solution for governmental agencies tasked with
sealing orphan wells under the BIL. Future developments in sensor optimization, including the use of more affordable wind
and methane detectors, are expected to further enhance its deployment efficiency and accuracy across diverse field
conditions. Further testing is being done to optimize the necessary wind and methane sensors to lower costs and maintain
accuracy in order to deploy this technology across the U.S. to quantify fugitive emissions and mitigate their near-term
impacts of climate change.

**Appendix A: Comparison to Gaussian Plume Method**

The approach to deriving the equations governing the FAST method outlined in the "Mathematical Model" section
can also be compared to the more traditional approach using a Gaussian Plume model (GPM). Through this comparison, we
can gain deeper insight into the physical significance of the proportionality constant ($\beta$), as it relates to the diffusivity of the
pollutant of interest. Including a term for reflection from the ground (but not from an inversion aloft), the GPM estimates the
downwind concentration of a pollutant as a function of the emission rate (Q), advective velocity (u), crosswind distance from
centerline (y), vertical displacement from centerline (z), height of the emission source (H) and horizontal and vertical
dispersion coefficients ($\sigma_y$ $\sigma_z$) as follows:



$$C(x,y,z) = \frac{Q}{2\pi\,\sigma_y\,\sigma_z\,u}\,exp\{\frac{-y^2}{2\sigma_y{}^2}\}(exp\{\frac{-(z-H)^2}{2\sigma_z{}^2}\} + exp\{\frac{-(z+H)^2}{2\sigma_z{}^2}\}) \tag{A1}$$

Similar to the FAST model, the GPM assumes that the velocity profile is constant in space. However, the GPM does not assume the concentration profile is constant, which is implicitly done by the FAST method via the use of centerline time-averaged concentrations (Figure 1A). Rather, the GPM assumes that the concentration profiles are Gaussian in the y and z directions, with standard deviations ($\sigma_y$, $\sigma_z$) related to the width of the plume. These standard deviations are often approximated using empirical data (i.e. Pasquill stability classes) but can be defined exactly using the diffusivity of the

pollutant (D). Assuming that the plume is isotropic and homogeneous, we can define:

$$\sigma_y \approx \sigma_z \approx \hat{\sigma} = \sqrt{\frac{2Dx}{u}} \tag{A2}$$

where D is the diffusivity of the pollutant and $\hat{\sigma}$ is the standard deviation of the Gaussian plume in all directions orthogonal to x. Evaluating this equation at some downwind distance $x_0$ and substituting our earlier use of centerline velocity measurements (given that the velocity profile is assumed constant in both GPM and FAST), we can define the standard

deviation there ($\hat{\sigma}_0$):

$$\hat{\sigma}_0 \approx \sqrt{\frac{2Dx_0}{u_{CL}}} \tag{A3}$$

Using this standard deviation, we can imagine integrating the FAST approach over a certain number of standard deviations to capture more and more of the true concentration profile. To capture 99.7% of the total plume, we would need to integrate

out to three standard deviations, or $3\hat{\sigma}_0$. Using this comparison to the previous equation derived for FAST (Equation 2), we can solve for $\beta$.

$$\sigma_0 = \sqrt{\beta\,i_{fan}\,l_{fan}\,x_0} \simeq 3\hat{\sigma}_0 \approx 3\sqrt{\frac{2Dx_0}{u_{CL}}}$$
$$\beta \approx \frac{18\,D}{i_{fan}\,l_{fan}u_{CL}} \tag{A4}$$


Here, we find that the proportionality constant $\beta$ can be understood as a non-dimensional ratio of two diffusivities - one being the true diffusivity of the gas and the other being due to the turbulence generated by the fan. Since D can be very difficult to measure, the FAST method provides a work-around such that only constants related to the fan-generated flow need to be defined to quantify the emission rate. Equation 8 could also be inverted to estimate the diffusivity D, but this is

not of real interest for this study



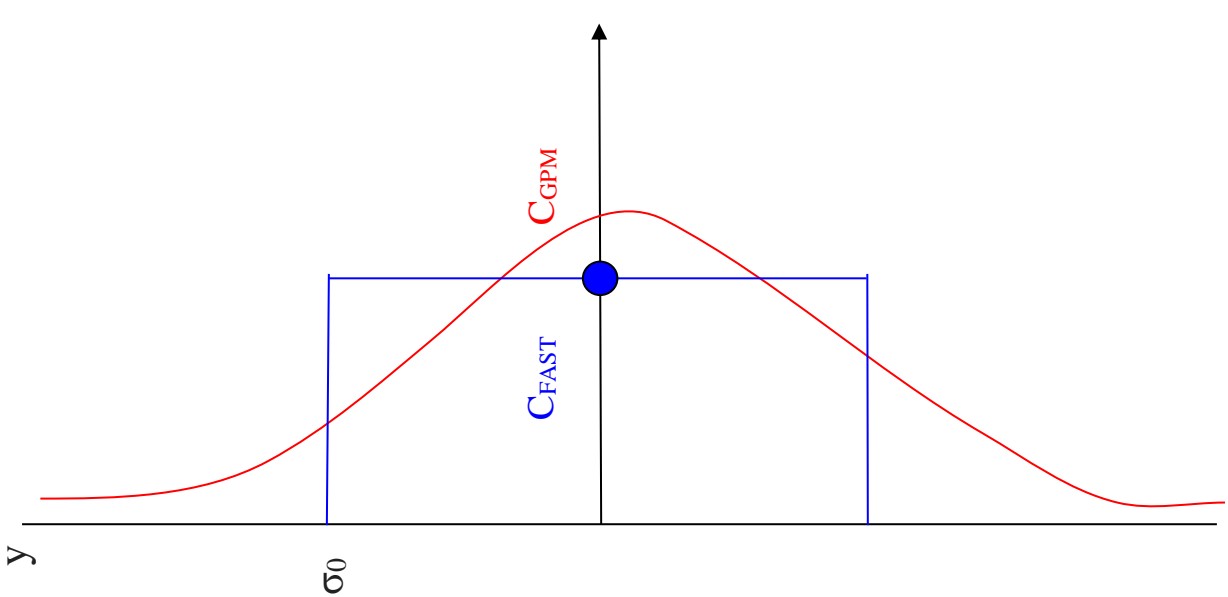

Figure 1A: Diagram showing the difference in concentration profiles (C) used by the FAST (blue) and GPM (red) methods

**Author contribution**:

Mohit L. Dubey: Conceptualization, Data curation, Formal analysis, Investigation, Methodology, Software, Validation, Visualization, Writing

Andre Santos: Conceptualization, Data curation, Investigation, Methodology, Validation, Visualization, Writing

Andrew B. Moyes: Conceptualization, Investigation, Methodology, Validation, Writing (review and editing)

Ken Reichl: Conceptualization, Data curation, Investigation, Methodology, Validation, Writing

James E. Lee: Investigation, Writing (review and editing)

Manvendra K. Dubey: Investigation, Writing (review and editing)

Corentin LeYhuelic: Investigation, Formal analysis, Writing (review and editing)

Evan Variano: Formal analysis

Emily Follansbee: Investigation, Writing (review and editing)

Fotini K. Chow: Conceptualization, Formal analysis, Writing (review and editing)

Sébastien C. Biraud: Conceptualization, Investigation, Methodology, Validation, Writing (review and editing), Project

administration, Funding acquisition





**Competing interests**:

The authors declare that they have no conflict of interest.

**Acknowledgements**

This work was supported as part of the Consortium Advancing Technology for Assessment of Lost Oil & Gas, funded by the U.S. Department of Energy, Office of Fossil Energy and Carbon Management, Office of Resource Sustainability, Methane Mitigation Technologies Division's, Undocumented Orphan Wells Program. This material is based upon work supported by the U.S. Department of Energy, Office of Science, Office of Advanced Scientific Computing Research, Department of
Energy Computational Science Graduate Fellowship under Award Number(s) Grant Number: DE-SC0024386)]. Authors at Lawrence Berkeley National Laboratory are supported under Contract No. DE-AC02-05CH11231 with the U.S. Department of Energy. The U.S. Government retains, and the publisher, by accepting the article for publication, acknowledges that the U.S. Government retains a non-exclusive, paid-up, irrevocable, world-wide license to publish or reproduce the published form of this manuscript, or allow others to do so, for U.S. Government purposes. We also thank the people of the Osage
Nation, Oklahoma, for providing access to their field sites and the US Forest Service for hosting the Lufkin, TX sampling.

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
