# Peer review of "Development of a Forced Advection Sampling Technique (FAST) for Quantification of Methane Emissions from Orphaned Wells"

_EGUsphere, 2024_

## Author Comment (AC3)

**Reviewer 1**

Review of "Development of a Forced Advection Sampling Technique (FAST) for Quantification of Methane Emissions from Orphaned Wells"

October 22nd 2024

This paper describes a novel approach to a current hot topic in science.  The authors describe a new method to quantify methane emissions from orphaned oil and gas wells.   Controlled release experiments (~1 g CH4 h-1 to 40 g CH4 h-1) were used to tune the model and then were compared against another instrument's data.  The FAST system then is deployed in the field to measure emissions from four orphaned wells in a real-world setting.  The paper is well written but overlong in many places and could benefit from some text being removed completely and extraneous figures moved to the SI.  One major concern I have is that the conclusion reads like a commercial advertisement for the FAST method and I would urge caution about overselling this method's ability and making claims about the efficacy of other technologies/approaches that have been validated.

This study describes an in interesting idea, but I have many concerns about the methodology and the main conclusion that this method is a "low-cost, portable, fast and safe alternative to existing methods" is more subjective rather than what is shown by the data.  I do think that the FAST method could be used in special circumstances that other methods cannot, however, I am not encouraged to adopt the FAST method for abandoned well measurement.  My general concerns on the methods, results and conclusions are as follows with specific comments on the manuscript thereafter.

General comments

1. The main conclusion that the FAST method is lower-cost, more portable, faster and safer than existing methods appears to come from arbitrary comparison presented in Table 1. The data presented in Table 1 is at best misleading and in some cases incorrect. This table should be improved by using quantitative variables instead of the undefined "Accuracy", "Size", "Labor" and "Safety".  For "Accuracy" you could use the "Uncertainty"; "Size" present the volume of the measurement equipment with appropriate units; "Labor" you could use time taken for 1 measurement (including set up and taking the measurement); and "Safety" a binomial of whether an operator is likely to be exposed to unprocessed natural gas.  Three key variables missing here are "Will it work in a complex aerodynamic environment (i.e. wells in woodland)" and "Does it account for undetected sources", and "Does environmental variability affect precision/accuracy?".

Please also review the "Hardware" cost data as these are incorrect.  You can build a reasonably good static or dynamic chamber for $400 and the FAST system comprising of a Picarro GasScouter, Sonic anemometer and multiple laptops would cost >$50k (if the Picarro GasScouter was still available to buy).  The static/dynamic chambers were initially used to

measure emissions from abandoned wells as the researchers did not have access to trace gas analyzers and were encouraged to develop lower cost/tech methods of quantification.

My expectations are that with a correct Table 1, the FAST method (as described in the methods section) would be relatively expensive, difficult to set up, difficult to transport but could be considered safer than many approaches, will work in complex aerodynamic environments, and could be used to quantify emissions from a larger piece of infrastructure that has many leaks. This method could be used for measuring emissions from abandoned pump jack wells that won't fit in a chamber and are surrounded by trees and suggest the authors focus on realistically evaluating the FAST systems performance and utility in the real world.

We agree that Table 1 had errors that would be misleading to readers. We have updated Table 1 to include specific variables that are more quantitative (setup/measuring time and size) while keeping the values somewhat approximate and included a row for "versatility" based on the ideas of this review. We also updated the caption to more specifically reflect the uncertainty in these values and the caveats related to the FAST method.

| Method | FLIR Camera | SEMTECH HI-Flow 2 | Static Chamber | Dynamic Chamber | GPM | Vent | UAV | OTM | FAST |
|---|---|---|---|---|---|---|---|---|---|
| **Hardware Cost** | >$50K | ~$40K | >$400 | >$400 | >$5K | ~$50K | >$50K | >$10K | $2K-50K* |
| **Range (g/h)** | >100 | <1-30,000 | >0.1 | >0.1 | >100 | >100 | >50 | >50 | >1 |
| **Uncertainty** | High | Low | Low | Low | High | Low | High | Low | High |
| **Size (L)** | ~0.3 | ~15 | ~20 | ~20 | ~50 | N/A | ~40 | >1,000 | ~10-50** |
| **Measuring Time (min.)** | ~2 | ~3 | >30 | >30 | >10 | >30 | >30 | >10 | ~3 |
| **Setup Time (min.)** | ~5 | ~5 | >10 | >10 | >10 | >10 | >30 | >30 | ~5-30** |
| **Safety*** | High | Low | Low | Low | High | Low | High | High | High |
| **Versatility**** | High | Low | Low | Low | Low | High | Low | Low | High |

**Table 1**: Comparative assessment of commercial (FLIR, SEMTECH, Vent) and research (Chamber, GPM, UAV, OTM) methods used to monitor fugitive methane leaks from orphaned wells. Hardware costs, detection range, accuracy, size, labor and safety are compared for each technology. *The FAST method costs are currently limited by the high cost of laser trace gas sensors (Picarro, Aeris, etc.) that can be reduced significantly by using cheaper non-laser sensors (i.e. Gas Rover) used in chambers. **The size and setup time of the FAST method can also be decreased by using a leaf-blower type fan and a more compact sensor setup. ***Safety reflects the likelihood of an operator not being exposed to unprocessed natural gas.

****Versatility reflects the ability of the method to work in complex aerodynamic environments (i.e. wooded areas, remote areas) and on a wide range of well types.

2. The assumptions about abandoned wells are very simplified. In your study, the controlled releases assume that all emissions from abandoned wells are single, continuous point sources of between 0.9 g CH4 h-1 and 40 g CH4 h-1 in the middle of a grass field. This is not the case. I would have expected the controlled release experiments to simulate many more emission scenarios across the full range of emission rates.  The largest reported emission from and abandoned well was 70 kg CH4 h-1.  What is the lower limit of quantification of the FAST method, what is the highest?  What were the meteorological/micrometeorological conditions and did these affect quantification?  How is the experimental set up in Figure 8 different from what was encountered in the field (Figure 12)?  Many things are missing from if this is to be considered a validated quantification approach.  It would be acceptable to say that "The FAST method has been tuned using single, continuous point sources between 0.9 and 37 g CH4 h-1 at 1 m above the ground, in a simple aerodynamic landscape (grass field), in X, Y and Z meteorological conditions, and in low cross-wind conditions (< X m/s)."

 We agree that the assumptions made in this paper are generally very simplified. As I address below (Comment 3), the controlled release experiment was used to verify that the method (using the fan in ambient wind conditions) would provide meaningful results. However, the FAST method should be expanded upon and tested in a variety of wind conditions in order to be established as a vetted technology for measuring emissions. In this study, we were limited to the areas in which we had access (Richmond, CA, Osage, OK and Lufkin, TX) but aim to expand this to more diverse areas in future research.

Based on the study done in this paper we can say that the lower limit of quantification for the FAST method is around 1 g/hr as that is the lowest value we measured. We cannot explicitly say what the highest is, as we only measured up to 40 g/hr in our controlled release and our field measurements were also below 100 g/hr. We are currently doing more controlled release experiments to better understand the upper limit of the method, but they are beyond the scope of this paper. While meteorological conditions do appear to have some effect on the results of the FAST method, the method is far more robust to a wide range of wind conditions due to the strength of the fan. Note that, in Texas, we only used the fan on a "low" setting, as the background winds were low, while we used the fan on a "high" setting in Oklahoma to account for the elevated background winds. This can be seen as a strength of the method compared to Gaussian Plume approaches which rely on the ambient winds to generate the plume.

We have included many of the caveats addressed in this comment now in the revised manuscript and have been more careful in how we make claims about the FAST method.

3. The controlled experiments do not validate the approach, but instead are used to generate the fan specific value for KFAST. The way data are presented is a little misleading as the emission rates calculated (in Table 3) using the FAST approach are presented before the calculation of KFAST (Figure 11). The emission rates in Table 3 presumably use the calculated

value for KFAST.  This (KFAST), and all other data used to calculate the FAST emission rates, should be clearly stated in Table 3 and include values for all variables in Equations 3 and 4.  As the controlled release experiments were used to generate one of the key variables for the FAST approach, it cannot be claimed that the method has been validated against controlled release experiments.  This would have added much more to credibility to the FAST approach and give more confidence in the emission rates calculated in real world settings, especially given the claims in the conclusion.

The control release experiment was done primarily to verify if the FAST method would actually work under ambient wind conditions. The determination of the values of $K_{FAST}$ was simply an added benefit of doing the controlled release that allowed us to also determine uncertainties on the value of $K_{FAST}$ which were then used to better estimate the uncertainties of field measurements. In principle, the values of $K_{FAST}$ can be simply derived from Equation 4:

$$K_{FAST} = \pi \, \sigma_0^{\,2} = \pi \, \beta \, i_{fan} \, l_{fan} \, x_0$$

One simply needs to measure the blade length ($l_{fan}$) and turbulence intensity ($i_{fan}$) of the fan, both of which can be done without a controlled release experiment. Using the turbulence intensity from our preliminary fan studies with $\beta = 1$, $i_{fan} = 0.23$, $l_{fan} = 0.13 \, m$, $x_0 = 2 \, m$, the

effective $K_{FAST}$ would be $K_{FAST} = \pi \, \beta \, i_{fan} \, l_{fan} \, x_0 = 0.19$. This value matches the value determined during the controlled release experiment with maximum filtering (300 degrees), where both the low and high fan settings agree.

Using this value of $K_{FAST}$ as well as the standard deviations of measured wind speed and methane concentration to estimate the uncertainty, we rederive Table 3 below.

| Source Rate $Q \pm \sigma_Q$ (g/hr $CH_4$) | SEMTECH (g/hr $CH_4$) | FAST (No Fan) (g/hr $CH_4$) | FAST (Low Fan) (g/hr $CH_4$) | FAST (High Fan) (g/hr $CH_4$) |
|---|---|---|---|---|
| 0.93 ± 0.03 | 0.96 ± 0.03 | 0.27 ± 0.50 | 0.61 ± 0.49 | 1.01 ± 0.50 |
| 1.86 ± 0.06 | 1.89 ± 0.05 | 0.23 ± 0.65 | 1.76 ± 1.11 | 2.81 ± 0.78 |
| 4.66 ± 0.17 | 4.62 ± 0.08 | 1.41 ± 2.76 | 3.56 ± 2.38 | 4.65 ± 2.38 |
| 9.33 ± 0.32 | 9.25 ± 0.16 | 0.41 ± 1.99 | 9.77 ± 4.74 | 9.75 ± 4.08 |
| 18.67 ± 0.63 | 18.30 ± 0.33 | 0.09 ± 0.34 | 15.90 ± 9.21 | 22.80 ± 7.14 |
| 37.32 ± 1.27 | 36.90 ± 0.53 | 5.57 ± 13.00 | 38.12 ± 20.62 | 38.20 ± 17.40 |

This updated table (in lieu of Table 3 removed) validates that the expected value of $K_{FAST}$ from Equation 4 is useful to predict the controlled release flow rate for both the low and high fan speeds with the data filtered by an angle of 300 (ignoring out of plume data).

We have also added a table to the body of the paper which contains all relevant parameters used to calculate the emissions rates in our field campaigns.

|  | No Fan | Low Fan | High Fan |
|---|---|---|---|
| **0 Degrees** | 1.70 ± 0.52 | 0.47 ± 0.03 | 0.27 ± 0.08 |
| **180 Degrees** | 1.63 ± 0.51 | 0.28 ± 0.01 | 0.24 ± 0.03 |
| **300 Degrees** | 1.11 ± 0.37 | 0.20 ± 0.01 | 0.19 ± 0.01 |

Table 3: Values of $K_{FAST}$ and their associated uncertainties under various filter and fan conditions determined from the Richmond controlled release experiment.

4. The results of the "Hooper #41" well are very telling. This is a well that looks like it has been capped and cut near the surface and is leaking, there are many of these across the country. From Figure 12, it is very likely that the fan is moving the air above the well head towards the sensor inlet and it is unlikely that all emitted gas will be entrained in the air flow. I suspect this could result in lower concentrations being detected and lead to an underestimation of emission (as is shown in Section 2.3.2.2 and Figure 16). This is strongly suggested by the data from the SEMTECH.

This suggests that there could be a major shortcoming in the methodology, i.e. when the fan does not blow gas from the point of emission to the analyzer inlet. This makes measuring abandoned wells that are effectively holes in ground very difficult. The confirmation of this could be proven using controlled release experiments at different heights (down to 0 m AGL), I do not expect you to do this, but this key shortcoming should be discussed in the paper.

This is an interesting and perceptive comment, as we had not considered exactly why the readings from the SEMTECH and FAST method were so different in the case of Hooper #41. We have updated the section on Hooper #41 to include the following discussion:

"The results for Hooper #41 highlight challenges in measuring methane emissions from variable wells and suggest potential limitations in the FAST method. The variable leak rate led to significant uncertainty in SEMTECH readings (±95.47 g/h), while the FAST method provided more stable estimates (10 ± 10 g/h). However, the fan setup likely failed to fully entrain the emitted gas into the airflow directed toward the sensor, potentially leading to an underestimation of emissions, as supported by SEMTECH data and Figure 16. This limitation is particularly critical for wells with low-height emissions, such as Hooper #41. Despite this, the FAST method shows promise for measuring variable emissions more consistently than SEMTECH. Future

work could address this limitation through controlled release experiments at different heights to optimize the fan and sensor configurations for capturing low-lying plumes."

Further discussion about Hooper #41 in other sections of the paper were modified to incorporate this uncertainty.

5. The assumption that σ0, as defined in Halloran et al (2014), can be used in this context is another key assumption of the approach. Halloran et al. (2014) used smoking oil to generate their results while this study is modelling the dispersion of methane emitted at pressure through a small aperture. Some comment of this is warranted in the discussion.

Agreed, we have added the following paragraph to address this comment:
A key assumption of this study is that the effective plume width $(\sigma_0)$ derived in Halloran et al.

(2014) for smoking oil plumes is applicable to methane dispersion from orphaned wells. While the MDB fan generates turbulent transport similar to Halloran et al., differences in the physical properties of smoking oil and methane—such as buoyancy, diffusion rates, and emission dynamics—could lead to deviations in plume behavior. These potential differences underscore the need for additional experiments designed specifically for methane to validate the use of $(\sigma_0)$ under these conditions and further refine the FAST method's applicability.

Specific comments

Abstract

L12: This is the definition for abandoned wells.  Not sure how an environmental challenge can be both "significant" and "poorly understood".

Updated to include more specificity per  this definition: "Orphaned oil and gas wells are unplugged nonproducing wells with no solvent owner of record to plug and mitigate them, such that the responsibility often falls on government agencies and the general public." (https://pubs.acs.org/doi/10.1021/acs.est.2c03268)

"Poorly understood" has been changed to "undersampled".

L14: As stated above the FAST method doesn't really improve on current methods, it provides an alternative measurement method.

Updated to "alternative" to reflect ambiguity.

L15: Is the flow of air a "jet"?
"Jet" updated to "force advection".

L20: Conclusion isn't well supported by the rest of the paper. Caveats to the FAST method's performance should be stated here.

Added "below 40 g/h and within certain geometric and atmospheric constraints."

Introduction

L24: Repeat of L12. Same issues.

Addressed in the same fashion.

L26: EPA estimates there are ~3.5 million abandoned wells in the US.

Yes, but here we are only concerned with orphaned wells (see IOGCC 2021 report) which are a subset of abandoned wells. EPA estimate can be found here: https://www.doi.gov/sites/default/files/federal-orphaned-wells-methane-measurement-guidelines-final-for-posting-v2.pdf

L27: "imagined" sounds like the wrong word.

Changed to "thought"

L30: Can you explain methane is an issue.

I'm not sure I understand what you are asking here.

L31-33: Lines starting "Furthermore…", who is underestimating emission and by how much. Is this total national emission estimates? If so, how does this relate to the total emissions from the country? If there is a large underestimation from a large source, this is a problem. If the underestimation is from a small source, it is likely to be in the noise of the total national emission and not a significant national problem. Need to be careful here about potentially inflating the size of the problem caused by orphaned wells. Are there any papers out there that suggest total regional emissions from abandoned wells are a representative proportion of total regional emissions?

This is a direct reference to Williams et al. 2021 which used a small dataset of orphan wells to determine that "annual methane emissions from abandoned wells are underestimated by 150% in Canada and by 20% in the U.S." We are not making any inflated claims about the total size of orphaned well emissions, just that they are significantly undersampled. We removed any use of the word "underestimated" from our text.

L33: "measuring" instead of "plugging"?

Updated to "measuring and plugging"

L36: Could add "measuring and plugging orphaned wells to achieve climate goals."

See above.

L41: Why such the large range? Did this change over time or is it regional?

Good question! Likely due to the fact that some wells are very remote and expensive to fix, while others are more accessible.

L51: What proportion of wells are > 1 kg/h?
Addressed in next sentence:
"extremely high-emitting orphaned wells are very rare"

L52: Double "]]"

Fixed.

L53-55: Not strictly true, Bridger Photonics claim a lower quantification limit of 1 kg/h. If all the wells emitting > 1 kg/h were plugged, what fraction of total emissions would be removed. This is the argument for all the remote sensing companies, get rid of the long-tail and the problem will be solved.

True, however, based on previous study about orphaned wells, specifically, most of the wells are far below this quantification limit. It is hard to determine what fraction of the emissions would be removed since so few orphaned wells have been sampled.

L59-60" "Unmanned…" what vehicles and why are they promising?

Added "also known as drones". Promising because they can cover a large area more easily than human/vehicle surveys.

L60: As mentioned above, the hardware costs in the table are incorrect. This is a key selling point of the FAST method and it is based on flawed data. Please revise the cost in the table and revisit the text.
Agreed, see response to Comment 1.

L64-70: As detailed in general comment 1 above, the table is too qualitative and does not contain metrics that can be reasonably compared. This is key to why a reader would choose the FAST method over the others, currently the reasoning is flawed.

Agreed, see response to Comment 1.

L83: Again, not sure "jet" is the correct word.

Fixed again.

L83: The FAST method requires a sonic anemometer, therefore, it is measuring the atmospheric stability.  Why can this not be used in the GPM?

It can be. This comparison between a FAST-like GPM and our work is outlined in the Appendix.

L86: Concluding line sentence should contain the caveats described in general comment 2.

Added "potentially" and "under reasonable meteorological conditions" to reinforce the caveats.

L90-95:  This is incorrect.  There are many studies that have used a GPM to calculate emission from less than 1 km.  Some studies have even validated the use of the model using controlled releases and generated uncertainty bounds.  It would be useful to include these studies in your paper.

Good point, I have updated the text to reflect this and cite this paper which validated AERMOD (GPM based) to ~50m. https://doi.org/10.1175/JAM2228.1

Methods

L145: Add location of manufacturer.

Added "United Kingdom"

L146: Add detail of the duct blaster.  Why was this specific fan chosen?  Was it the same type as in Halloran et al?

Not the same fan as Halloran et al. but similar. Clarified in text: "This fan was chosen as it is similar to those used in Halloran et al. and can be easily operated in the field at multiple fan speeds controlled by a dial."

L155-205:  These are all results and should not be in a methods section.  This section is too long and should be condensed to important outcomes:  High setting should be used; measurements should be taken along the plume center line; and measurements should be taken less than 2 m from the fan.  Could put all the details and extra figures in the SI?

This has been reworked to be section 3.1 "Fan Characterization Results". Most of the text and figure have been moved to Appendix C: Fan Characterization Results, while the body of the text simply contains the relevant summary information.

L209:  Why was the SEMTECH Hi Flow used?

Clarification: "The SEMTECH HI-FLOW backpack system was used for verification as it has already been validated as a commercial product for estimating leak rates."

L234: Figure 7 should go in the results section, but it is actually unnecessary and should be removed or put in SI.

Figure 7 has been moved to the Appendix.

L242: Which MFC was used?  Add manufacturer and specs.

Brooks Instrument GF40, added.

L243: Add manufacturer details.

Added citation: *GasScouterTM G4302 Mobile Gas Concentration Analyzer | Picarro*. https://www.picarro.com/environmental/products/gasscoutertm_g4302_mobile_gas_concentration_analyzer. Accessed 7 Jan. 2025.

L244: How far?
Clarified, (inlet tube mounted within 1 cm of center of anemometer, see Figure 8)

L255: Why test the FAST approach without a fan?

This is to test the null hypothesis. If the method works without a fan, then it would be much simpler. However, we cannot know that the fan improves the results a priori without also testing it under the same conditions without the fan.

L260: A schematic would be better than a photograph.

Agreed, but the photograph is much easier to visualize exactly how the test was done. See response to comment above which references this photograph.

L308-341:  This is all results, not the method.  Please move to the Results section.

Moved to results (3.2 Experimental Determination of $K_{FAST}$).

Table 3: The uncertainties for the FAST method are very large +150%, -100% in some cases. Are these correct?  Why present data with no fan, it's not part of the method for the FAST approach and doesn't add anything here.  Also, don't need both Table 3 and Figure 10 – they are showing the same data.

This is a good point. As addressed above, we tested without a fan to ensure that the method improved with the fan (see response to L255). As it turns out, these values and their associated errors were not calculated correctly. The updated Table 3, which will be included in the Appendix of the paper is shown in the response to Comment 3. Here, the FAST errors are usually around 50% of the estimated value, which is similar to the observed uncertainty in the field. Figure 10 will be removed from the paper entirely as it does not add any beneficial information.

L348-359: These are also results.  Please move to the correct section.

Moved to results (3.2 Experimental Determination of $K_{FAST}$).

L377-384:  Superfluous detail, should be removed.

Agreed, shortened to say: "The first field campaign that measured orphaned wells using the FAST method took place in February 2024 in collaboration with multiple agencies. The U.S. Forest Service (USFS) invited the U.S. Department of Energy's Consortium Advancing Technology for Assessment of Lost Oil and Gas Wells (CATALOG) team to help measure and assess emissions from certain wells being plugged using funds from the Bipartisan Infrastructure Law (BIL). The FAST method was deployed in the field campaign to understand emission patterns better and help allocate sealing funds more efficiently."

Figure 13: Not necessary in the main body of the paper.  Should be moved to SI.

Moved.

L396: How was the leak detected?

Added clarification: "by sniffing the casing of the well with the Picarro G4302"

L427-430: Superfluous detail, should be removed.

Agreed, removed.

L435-439: Superfluous detail, should be removed.

Agreed, removed.

Figure 14: These are results and should be in the correct section.  Also, why are the error bars much smaller than those presented in the controlled release experiments?

The figure has been moved to results. The error bars here are smaller as there was less crosswind interference during this field experiment compared to the controlled release. However, as noted above, the controlled release values (and associated) errors were incorrectly calculated at first and have been updated, resulting in overall lower values.

L451-466: This measurement suggests fundamental flaws in the methodology. See general comment 3 above.

This has been addressed - see response to comment 4 above (not comment 3).

Results

L480: A very doubtful assumption.

Removed.

L505: The sentence starting "This shows…" is deeply flawed and should be rewritten.

Rewritten: "This shows that the measuring time for the FAST method, even at the Low Fan measurement, is similar to that of the SEMTECH (on the order of 3-5 minutes)."

L509-511: Please provide details of cost. I am not convinced by the qualitative sentence.

"The total cost of the sensors used in this study, a Picarro G4302 for concentration (~40,000 USD) and Gill Windmaster 1210-PK-085 (~5,000 USD) for wind, are about the same cost as a SEMTECH HI-FlOW (~50,000 USD)."

L513-518: Please provide more details of which sensors could replace the Picarro at a lower cost. The accuracy of the emission quantified by the FAST system are directly linked to the precision and accuracy of the analyzer taking the measurement, therefore, if the analyzer is replaced then the results presented in this paper will not be relevant to that system.

"Furthermore, the type of methane sensor can be optimized to a more reasonable price point, as CH4 signals near sources are typically high (e.g., > 1 ppm for leaks > 1 g/h). These levels can be effectively measured using less sensitive and more affordable technologies, such as near-infrared (NIR) sensors, non-dispersive infrared (NDIR) sensors, or solid-state absorption methods. For example, the MP400 series NDIR Dual Range Methane Sensor [MP400 Series NDIR], available for approximately $880, offers dual-range measurement capabilities (1-100% LEL/1-100% Vol), making it a cost-effective option. In contrast, the parts-per-billion sensitivity Picarro G4302, while highly precise, has limitations for the FAST method due to its low range, saturating above 800 ppm in methane mode. Future work will focus on investigating a wide variety of methane detection technologies to identify more cost-effective and reliable solutions for wide-scale FAST method deployment. Promising alternatives include portable devices such as the Bascom-Turner Gas-Rover II, designed for efficient leak surveys and priced at approximately $4,400 [*VGI-201/211 Gas-Rover II*], or advanced methods like Off-Axis Integrated Cavity Output Spectroscopy (ICOS) instruments, such as Nikira Labs' Portable Methane Gas Analyzer, which combines sensitivity and portability [Portable Methane Analyzer]."

"MP400 Series NDIR Dual Range Methane 1-100 %LEL/1-100 %Vol Sensor." *MPower Electronics, Inc*, https://shop.mpowerinc.com/products/mp400-series-ndir-dual-range-methane-1-100-lel-1-100 -vol-sensor. Accessed 9 Jan. 2025.

"Portable Methane Analyzer." *Nikira Labs*, https://www.nikiralabs.com/portable-methane-gas-analyzer. Accessed 9 Jan. 2025.

*VGI-201/211 Gas-Rover II Natural Gas Detector [2022] - $4,400.00 : Bascom-Turner Store - Portable Gas Detectors*. https://www.bascom-turner.com/store/index.php?main_page=product_info&products_id=462. Accessed 9 Jan. 2025.

L522-525: This sentence is incorrect.

Removed.

L528:  What I would like to see added here is the time taken to set up the FAST system and take one measurement as compared to the SEMTECH.

Updated: "Furthermore, while our existing proof-of-concept FAST hardware is currently heavier and more complex to operate than a SEMTECH, it could be replicated with a battery powered fan mounted to a tripod or a backpack vacuum blower, making it very similar to the size and labor requirements of the SEMTECH. Although both the FAST method and the SEMTECH take approximately three minutes to obtain a measurement, the FAST method currently requires a longer setup time (~30 minutes) compared to the SEMTECH (~5 minutes). However, with the aforementioned simplified setup, the FAST method's setup time could be reduced to match that of the SEMTECH, making it more practical for field deployment."

Summary and Conclusion:  This section makes very bold statements about the FAST method and the limitations of other instrumentation, some of which has not been borne out by the results.  Care should be taken here on the strength of the claims and the section should be rewritten from a more balanced point of view.

Updated:

"We have shown that using a fan-generated flow (forced advection) to create a jet between the

emission source and a point methane ($CH_4$) sensor and measuring 3D wind profiles using a sonic anemometer and $CH_4$ concentration with a gas analyzer (sampling technique), a simple estimate of the $CH_4$ emission rate of the source can be inferred (FAST method). The FAST method has been tuned using single, continuous point sources between 0.9 and 37 g/h at 1 m above the ground, in a simple aerodynamic landscape (grass field), in moderate meteorological conditions (< 5 m/s). Under these conditions, the FAST method consistently provided reasonable estimates of leak rates when fan speeds and filtering were applied appropriately, performing similarly to the commercially available methods (SEMTECH) and outperforming others (FLIR). Notably, the method's performance improves with increased fan speed and filtering angle. For instance, in the case of Rayburn #7 in Lufkin, Texas, the FAST method at the Low Fan speed consistently produced leak rate estimates that were within the uncertainty bounds of the SEMTECH values after just a few minutes of measurement. Without the use of a fan, the results showed much greater uncertainty, highlighting the importance of airflow in stabilizing methane dispersion for accurate estimation.

In the Texas and Oklahoma field campaigns, the FAST method was able to provide accurate and rapid readings under varying environmental conditions with errors on the order of 95% of the emission rate across a variety of wind conditions and leak rates. In Texas, where wind speeds were low, only the Low Fan setting was used, and FAST results aligned closely with SEMTECH, within 10%. Higher wind conditions in Oklahoma, however, required both Low and High Fan settings to account for greater natural dispersion. In the case of the highly variable Hooper #41, where SEMTECH struggled with the well's fluctuating leak rates, the FAST method's larger sampling cross-section and volume resulted in overall lower emissions estimates and relative uncertainty. However, the fan-driven airflow may not fully entrain all emitted gas, particularly from low-height leaks, potentially leading to an underestimation of emission rates for such wells.

The FAST method could provide a cost-effective, scalable, and practical alternative to existing technologies such as SEMTECH and FLIR for identifying high-priority orphan wells. The combination of fan-induced plume dispersion and real-time methane measurement allows for quick assessments (on the order of minutes), making it well-suited for large-scale monitoring. Future developments in sensor optimization, including the use of more affordable wind and methane detectors, are expected to further enhance its deployment efficiency and accuracy across diverse field conditions. Further testing is being done to optimize the necessary wind and methane sensors to lower costs and maintain accuracy in order to deploy this technology across the U.S. to quantify fugitive emissions."

L565: What I would like to see is an analysis done using GPM using the data from the "no fan" scenario.  How does this compare to the FAST method emission estimates?

We calculated the estimated emissions from a Gaussian Plume Model (GPM) using three different Pasquill Stability Classes (A, B and C) based on the no fan concentration and wind measurements for each well (shown in the table below). As evidenced by the order of magnitude range for each well, the GPM is highly sensitive to the choice of stability class, which

is not immediately apparent for such short range measurements. We used the equations for calculating $\sigma_y$ and $\sigma_z$ from Cooper, C. D. and Alley, F. C. Air Pollution Control: A Design Approach, Edition 4, Waveland Press, Inc., Long Grove, IL, 2011, pp. 662-663.

| Date | Well ID | SEMTECH (g/hr) | FAST (300 Filter) (g/hr) | GPM for Daytime Pasquill Stability Classes (g/hr) |
|---|---|---|---|---|
| 2024-02-06 | Rayburn #7 (Lufkin) | 2.9 ± 0.0 | Low: 2.6 ± 1.9 No: 0.6 ± 1.8 | Class A: 3.41 ± 20.2 Class B: 0.87 ± 5.18 Class C: 0.01 ± 0.08 |
| 2024-02-07 | Undisclosed Well (Lufkin) | 1.0 ± 0.3 | Low: 0.5 ± 0.7 No: 1.3 ± 3.4 | Class A: 1.05 ± 13.5 Class B: 0.27 ± 3.45 Class C: 0.0 ± 0.0 |
| 2024-03-14 | Humphrey #5 (Barnsdall) | 2.0 ± 0.04 | High: 2.0 ± 1.1 Low: 2.5 ± 1.5 No: 10.0 ± 15.1 | Class A: 226 ± 336 Class B: 57.8 ± 85.8 Class C: 1.06 ± 1.57 |
| 2024-03-14 | Hooper #41 (Barnsdall) | 70.1 ± 95.5 | High: 9.3 ± 10.4 Low: 9.9 ± 12.7 No: 1.0 ± 4.0 | Class A: 70.0 ± 332 Class B: 17.8 ± 84.2 Class C: 0.0 ± 0.0 |

Based on there being moderate to strong insolation and low wind speeds (<2 m/s) for the Texas wells, the most likely stability class for Rayburn #7 and Undisclosed Well is Class A or B. Using these stability classes, the GPM does reasonably well at estimating the magnitude of the leak but the uncertainty is much higher than both the SEMTECH and the FAST method. Due to the higher background wind speeds (3-5 m/s) and moderate to strong insolation in Oklahoma, the B and C stability classes are most likely for Humphrey #5 and Hooper #41. Here, the GPM overestimates the magnitude of Humphrey #5 by orders of magnitude and the uncertainty is very high. For Hooper #41, which is a highly variable well, the GPM also still performs poorly relative to the SEMTECH and FAST methods.

This discussion is included in Appendix B.

---

## Author Comment (AC4)

**Reviewer 2**

Review of "Development of a Forced Advection Sampling Technique (FAST) for Quantification of Methane Emissions from Orphaned Wells"

December 24th 2024

Dubey et al. make measurements of methane emissions from orphaned wells using a new Forced Advection Sampling Technique (FAST). This technique places a fan upwind of a suspected point source, which blows the emissions toward a measurement location. Using measurements of the wind speed and some plume shape assumptions, they calculate an emissions rate. Their technique is compared with results from a SEMTECH HI-FLOW backpack in a controlled release experiment, and then used to determine emission rates from 4 real-world orphaned wells.

This paper investigated a new technique for determining natural gas emissions from orphaned wells. As such, it is worthy of publication. However, this reviewer would like to see some extra information in the discussion and results before I think it's ready for publication.

Some discussion of the filter angle needs to be included in the methods section of the text. It is hinted at in Figure 5, but never really discussed. In Figure 4, the caption says data were filtered to remove points coming from the negative x direction (180 degrees). How many points were these relative to the total? Later, a filter of 300 degrees is used, but I don't know what that means. If 180 degrees is the negative x direction, I am guessing a crosswind be either 90 or 270 degrees? In line 352, the authors state the crosswind is filtered as the filter angle approaches 360. Is that the same as saying it approaches 0? Or in line 356, when the authors state a filter angle of 300 filters out all crosswind interference, does this mean they are only looking at 60 to -60 degree wind directions? Or from 120 to -120? Since so much of the discussion (Fig. 16, Table 4, etc.) depends on understanding this filtering, I don't think I can give a full final review of this manuscript.

Thank you for your very insightful comments and review. The lack of discussion of our filtering methods was a pretty glaring hole in the methodology and we have included an entire new section of the paper to address this, see below.

**2.2.3 Data Filtering by Wind Direction**

In order to optimize the FAST method under strong crosswind conditions, filtering was applied to improve data quality and estimate emissions more accurately. Despite the advection from the fan, strong crosswind interference (where $v > u$) introduces variability in both the concentration (C) and wind speed (u) measurements. Filtering addressed this issue by excluding data associated with wind directions unlikely to transport emissions directly to the sensors.

To filter the data, we first calculate the wind direction ($\Theta_i$) from the x- and y-direction wind components (u and v) within a normalized range of [0, 360) degrees for each data point as follows:

$$1)\quad \Theta_i = ((arctan2(v, u) * \frac{180}{\pi}) + 360)\ mod\ 360$$

The mean wind direction, $\Theta_{mean}$, is then computed as the arithmetic average of the normalized wind directions:

$$2)\quad \Theta_{mean} = \frac{1}{N}\sum_{i=1}^{N} \Theta_i$$

where N represents the total number of data points in a given measurement period.

We then apply a filter angle ($\phi$) symmetrically around the mean wind direction to define the range of included data. The lower ($\Theta_{lower}$) and upper ($\Theta_{upper}$) bounds of the filtered range are defined as:

$$3)\quad \Theta_{lower} = (\Theta_{mean} - \frac{360 - \phi}{2})\ mod\ 360$$

$$\Theta_{upper} = (\Theta_{mean} + \frac{360 - \phi}{2})\ mod\ 360$$

The wind and methane data are then filtered to include only directions within the specified range. If the bounds do not cross the 0°/360° discontinuity, the filtered data satisfies:

$$4)\quad \Theta_i > \Theta_{lower}\ \text{and}\ \Theta_i < \Theta_{upper}$$

and when the bounds span the discontinuity, data satisfying the following conditions are used:

$$5)\quad \Theta_i > \Theta_{lower}\ \text{or}\ \Theta_i < \Theta_{upper}$$

Figure 5 illustrates the impact of varying the filter angle on the time series of wind and methane concentration data, for a 1 g/hr release from the Richmond Field Station experiment (N = 300). As the filter angle decreases, more data from crosswind and background noise is excluded (shown in red) and the mean wind speed (u) and concentration (C) values change, resulting in different estimates from the FAST method. We found that a filter angle of 300° effectively aligns the analysis with wind directions closely aligned with the source when accounting for plume spread within x < 2 m.

[Figure]

**Figure 5**: Time series of wind speed (u) and methane enhancement (C) as well as C vs. wind direction (in degrees) for various filtering angles from a 'no fan' release of 1 g/hr at the Richmond Field Station. Kept data are shown in blue while filtered data are shown in red. Mean u and C over the 5 minute measurement period are shown in green.

It seems like one of the bigger uncertainties is what Figure 6 would look like for concentration in the y-z plane, especially for cases like Hooper #41. In fact, concentration or mixing ratio data are missing throughout the manuscript. Could the authors show concentration data for the reader to get an idea of how much variability and plume enhancement are involved in these calculations?

This is a very good point. Unfortunately, we did not have the time or equipment to measure these vertical profiles, but would like to do this in future studies to verify this plume behavior. In addition to the above figure we are also including a similar figure showing the concentration data for a single well (Rayburn #7) with and without the fan on to address this. See below:

[Figure]

Figure XX: Time series of wind speed (u) and methane enhancement (C) as well as C vs. wind direction (in degrees) for 'no fan' setting and various filtering angles from a 'low fan' setting. Kept data are shown in blue while filtered data are shown in red. Mean u and C over the 30 minute measurement periods are shown in green.

Figure XX illustrates the effect of using the fan on the time series of concentration and wind speed measurements at Rayburn #7, providing insight into the variability of methane concentrations and plume enhancements. Without the fan (upper left), the average wind speed in the x-direction (u) over the 30-minute measuring period was approximately 0 m/s. However, infrequent gusts in the x-direction caused spikes in methane concentrations, ranging from about 10 to 20 ppm above background levels. These spikes were spread across a wide range of directions, between 100 and 250°, indicating variable plume dispersion under stagnant conditions. With the fan on at a low setting, the mean wind speed in the x-direction (u) increased to approximately 2 m/s, and the plume became more stable. The methane concentration spikes were more concentrated in direction, between 180 and 210°, corresponding to the airflow from the fan. While a large spike was observed at the start of the low fan measurement, likely due to the fan turning on, the concentration stabilized to around 5 ppm above background levels.

Other comments:

I'm suspicious that the Williams et al. (2021) conclusion that a 20% uncertainty in orphaned well emissions is most uncertain emission in the U.S. Alvarez et al. (2018) estimated a 60% under-reporting from oil and natural gas production, for example.

We are quoting directly from Williams et al. (2021) abstract where they say: "We find that annual methane emissions from abandoned wells are underestimated by 150% in Canada and by 20% in the U.S. Even with the inclusion of two to three times more measurement data than used in current inventory estimates, we find that abandoned wells remain the most uncertain methane source in the U.S. and become the most uncertain source in Canada."

However, I agree that this claim is suspicious and vague. We have updated the sentence to be more simple:
'Based on a database of leak measurements at 598 wells across the U.S. and Canada, it was found that "annual methane emissions from abandoned wells are underestimated by 150% in Canada and by 20% in the U.S." [Williams et al., 2021].'

Table 1, why is the safety so low for the SEMTECH compared to FAST? I assume this is partly due to FAST diluting the plume quicker.

Yes - we have fully updated Table 1 per another reviewer's suggestions and included more clarification on the meaning of "Safety". See below:

| Method | FLIR Camera | SEMTECH HI-Flow 2 | Static Chamber | Dynamic Chamber | GPM | Vent | UAV | OTM | FAST |
|---|---|---|---|---|---|---|---|---|---|
| **Hardware Cost** | >$50K | ~$40K | >$400 | >$400 | >$5K | ~$50K | >$50K | >$10K | $2K-50K* |
| **Range (g/h)** | >100 | <1-30,000 | >0.1 | >0.1 | >100 | >100 | >50 | >50 | >1 |
| **Uncertainty** | High | Low | Low | Low | High | Low | High | Low | High |
| **Size (L)** | ~0.3 | ~15 | ~20 | ~20 | ~50 | N/A | ~40 | >1,000 | ~10-50** |
| **Measuring Time (min.)** | ~2 | ~3 | >30 | >30 | >10 | >30 | >30 | >10 | ~3 |
| **Setup Time (min.)** | ~5 | ~5 | >10 | >10 | >10 | >10 | >30 | >30 | ~5-30** |
| **Safety*** | High | Low | Low | Low | High | Low | High | High | High |
| **Versatility**** | High | Low | Low | Low | Low | High | Low | Low | High |

**Table 1**: Comparative assessment of commercial (FLIR, SEMTECH, Vent) and research (Chamber, GPM, UAV, OTM) methods used to monitor fugitive methane leaks from orphaned wells. Hardware costs, detection range, accuracy, size, labor and safety are compared for each technology. *The FAST method costs are currently limited by the high cost of laser trace gas sensors (Picarro, Aeris, etc.) that can be reduced significantly by using cheaper non-laser sensors  (i.e. Gas Rover) used in chambers. **The size and setup time of the FAST method can also be decreased by using a leaf-blower type fan and a more compact sensor setup. ***Safety reflects the likelihood of an operator not being exposed to unprocessed natural gas. ****Versatility reflects the ability of the method to work in complex aerodynamic environments (i.e. wooded areas, remote areas) and on a wide range of well types.

For Figures 2 and 3, is U defined in the x direction, or is it aligned to E-W?

U is defined in the x direction, added text to clarify: "For these experiments, u is aligned to be in the x direction (upwind/downwind), v in the y direction (crosswind) and w in the z direction (vertical)."

Figure 3, why is w so high?  Why wouldn't one expect that to be 0? Add discussion

Added clarification: "Moreover, the vertical velocity (w) is higher than expected for two main reasons: the anemometer is mounted at a height of 1 meter and the experiment was conducted on a rooftop. While w should be ~0 m/s at ground level, we measured w on the order of ~1 m/s due to these factors."

Figure 7, perhaps add the standard deviation of the emission rate to show "accuracy and precision … decrease" as stated in the caption.

The figure has been updated to include mean and standard deviation. This figure has also been moved to a new section in the appendix (Appendix B: SEMTECH Measurements) as they are not entirely relevant to the FAST method directly (per another reviewer's suggestion).

[Figure]

Line 402, How do the authors account for changing upwind background in an oil field? Is there variability in the background? This is one instance where concentration data would be helpful to get a sense for what is being measured. For example, what is the signal to background variability? Can the authors show a time series of measurements?

Added this to clarify: "At each well, we measured background (upwind) methane concentrations using the Picarro for five minutes and this background value was subtracted from the methane concentrations collected during the FAST method to determine the enhancement."

We now show time series of measurements in the new figure added to address filtering (Figure 5) and have added similar time series and filtering results for a well in Lufkin, TX (Rayburn #7), both shown above.

Fig. 15, would the SEMTECH estimate really be negative? Or is this a problem with an improper Gaussian distribution assumption?

Negative values would be non-physical, although the standard deviation reported by the SEMTECH does estimate this within the range of possible values. We have updated the figure to not allow negative values.

For Figure 1A, what is the blue dot?  Also, this graph seems to be limited by the resolution of the plume tests of Figure 6.  Wouldn't one still expect to have a Gaussian plume within the resolution of 0.3 meters?

The blue dot was unnecessary and has been removed. The point this figure is trying to convey is that while the Gaussian plume method (GPM) assumes a Gaussian plume shape, the FAST method collapses this Gaussian plume into an assumed average plume which is measured along the centerline and captures meaningful information out to some distance $\hat{\sigma}_0$.

Lines 58–59, please spell out all acronyms.

Updated to:
"... from expensive hand-held forward looking infrared cameras (FLIR) to more time-intensive mobile (OTM-33a) [U.S. EPA, 2014] and stationary systems (SEMTECH Hi-Flow 2 [SEMTECH], Chamber [Williams et al. 2023], Gaussian Plume Modeling (GPM) [Lushie and Stockie, 2010], Vent [Ventbusters, 2023]). Unmanned aerial vehicles (UAV, also known as "drones") have …"

SEMTECH is the name of a company and not an acronym.

---

## Referee Report (RR1)

Dear Authors,

Many thanks for addressing the previous comments. I am happy with most changes made to the specific comments but still have issues with some of the general comments.

General Comment 1

The table has been improved, however, there are still some issues:

- The uncertainty should be a value reflecting the percentage uncertainty of a single measurement (i.e. ±10%) and should be linked to a study that has measured this. This value is key to assessing the relative performance of methods.
- Please present the Hardware Cost of the FAST method as described in the paper and not a potential number based on sensors not used in this study. The value currently presented "$2k - $50k" is a range orders of magnitude apart and provides no useful information. Please only report the cost of the hardware as presented in this study.
- Again, size and set up time of the FAST method should reflect the system reported in this study and not a future untested system. How long did it actually take you to install a fan, sonic anemometer, Picarro etc and get them all running?
- I am a bit confused as to what "Safety" means as the caption reads "likelihood of an operator not being exposed to…". Does "low" mean a low likelihood of not being exposed, therefore a high likelihood of being exposed? Regardless, to me it seems that the GPM and FAST should have the same likelihood of being exposed to gas. Also, one key flaw of the FAST method is equipment needs to be installed upwind and downwind of the source, therefore, likely exposed to gas. In contrast, the Hi-Flow, OGI and Dynamic chamber can be installed from an upwind location and away from the plume. Therefore I would assume the FAST method is high risk while the Hi-Flow, OGI and Dynamic chamber are low risk.
- I also do not agree that the Hi-Flow has low versatility, this is a small, self-contained unit that has been used to measure methane emissions on the top of condensate tanks on production sites, so it can go almost anywhere. The results presented later in the paper suggest the fan has to be correctly positioned for the FAST method to be useful, therefore, it's not going to be very versatile for emission points that are hard to reach (most emissions on abandoned wells are difficult to reach).

I would suggest the following changes to the table

| Method | FLIR | Hi-Flow | Static chamber | Dynamic chamber | GPM | FAST |
|---|---|---|---|---|---|---|
| Cost | 50k | 40k | 400 | 400 | 40k | 50k |
| Range | Does not quantify | | | | | |
| Uncertainty | N/A | ±10% | -50%, +100% | ±15% | ±40% | ±50% |
| Size | 0.3 | 15 | 20 | 20 | 50 | 50 |
| Measurement time | 2 | 3 | 30 | 30 | 20 | 3 |
| Setup time | 5 | 5 | 10 | 10 | 10 | 30 |

| Risk of exposure | Medium | Medium | High | Medium | Low | Medium |
|---|---|---|---|---|---|---|
| Versatility | N/A | High | Low | Low | Medium | Medium |

General comment 2

Can you please share the caveats that you have added to the manuscript including the line numbers. As you have not included these details in the response I cannot comment if this issue has been addressed.

General comment 3

The addition of section 3.1 is good and clearly explains what was missing in the previous iteration. A necessary addition is the uncertainty of $K_{FAST}$, this is currently missing and essential to the study. Please add this.

General comment 4

The response is mostly OK apart from the statement "Despite this, the FAST method shows promise for measuring variable emissions more consistently than SEMTECH.". There is no evidence that the emission from Hooper #41 is constant and abandoned wells have shown to have variability on very short timescales. I suggest this sentence is removed.

General comment 5

Response is good.

Specific comments

All ok apart from those listed below.

Original L513-518 – Now P 25 starting "Furthermore, the type…"

This makes a very big assumption that lower cost sensors will have the same accuracy/precision as the Picarro and are sensitive enough to be used to measure ppm-level concentrations. In nearly all cases, this is not true. The NDIR and Gas-Rover will not be able to measure at low concentrations (< 10 ppm) while the Nikira Labs' Portable Methane Gas is still quite expensive (tens of thousands). This whole part of the discussion is highly speculative and should not be included, i.e. from "Furthermore," to "[Portable Methane Analyzer]."

L528

Again this is highly speculative. I would suggest the following statement is removed "However, with the aforementioned simplified setup, the FAST method's setup time could be reduced to match that of the SEMTECH, making it more practical for field deployment."

"5. Conclusions"

Several sentences within the "conclusions" section are not backed up by any of the findings in the paper. This section should be comprehensively reviewed.

For example, "In the case of the highly variable Hooper #41, where SEMTECH struggled with the well's fluctuating leak rates, the FAST method's larger sampling cross-section and volume resulted in overall lower emissions estimates and relative uncertainty. However, the fan-driven airflow may not fully entrain all emitted gas, particularly from low-height leaks, potentially leading to an underestimation of emission rates for such wells." How do you know the Hi-Flow "struggled"? This sentence should be rewritten.

The following sentence stating "Future developments in sensor optimization, including the use of more affordable wind and methane detectors, are expected to further enhance its deployment efficiency and accuracy across diverse field conditions. Further testing is being done to optimize the necessary wind and methane sensors to lower costs and maintain accuracy in order to deploy this technology across the U.S. to quantify fugitive emissions." I understand you are trying to convey that the system could be optimized to overcome some of the current shortcomings but it sounds like an advert for a product and shouldn't be included in a scientific paper as it stands. This should be rewritten.

---

## Author Response (AR2)

From Editor:

After reading the reviews, I have some additional thoughts as well.
1. Regarding the table of instrument comparisons, I strongly recommend removing any subjective rows, like "risk of exposure" or "versatility."

This has been done along with the suggestions of adjustments to Table 1 from Reviewer 1 (see below).

2. The purpose of AMT articles is to describe new instruments and measurements. While objective comparison to other methods is valuable, the focus should be on the technique/instrument itself, not how much better or worse it is than other methods (unless it were a true intercomparison study, which this is not). Removing some of the salesman-like language would make this paper more palatable for your readers. Let the numbers speak for themselves.

Agreed. All of the language of this nature has now been removed per the specific suggestions of the reviews below.

From Reviewer 1:

General Comment 1

The table has been improved, however, there are still some issues: • The uncertainty should be a value reflecting the percentage uncertainty of a single measurement (i.e. ±10%) and should be linked to a study that has measured this. This value is key to assessing the relative performance of methods. • Please present the Hardware Cost of the FAST method as described in the paper and not a potential number based on sensors not used in this study. The value currently presented "$2k - $50k" is a range orders of magnitude apart and provides no useful information. Please only report the cost of the hardware as presented in this study. • Again, size and set up time of the FAST method should reflect the system reported in this study and not a future untested system. How long did it actually take you to install a fan, sonic anemometer, Picarro etc and get them all running? • I am a bit confused as to what "Safety" means as the caption reads "likelihood of an operator not being exposed to…". Does "low" mean a low likelihood of not being exposed, therefore a high likelihood of being exposed? Regardless, to me it seems that the GPM and FAST should have the same likelihood of being exposed to gas. Also, one key flaw of the FAST method is equipment needs to be installed upwind and downwind of the source, therefore, likely exposed to gas. In contrast, the Hi-Flow, OGI and Dynamic chamber can be installed from an upwind location and away from the plume. Therefore I would assume the FAST method is high risk while the Hi-Flow, OGI and Dynamic chamber are low risk. • I also

do not agree that the Hi-Flow has low versatility, this is a small, self-contained unit that has been used to measure methane emissions on the top of condensate tanks on production sites, so it can go almost anywhere. The results presented later in the paper suggest the fan has to be correctly positioned for the FAST method to be useful, therefore, it's not going to be very versatile for emission points that are hard to reach (most emissions on abandoned wells are difficult to reach).

These are all very valid concerns for Table 1. The table has been updated and simplified to minimize confusion below. The non-quantitative rows were removed per the suggestion of the editor as well.

| Method | FLIR Camera | SEMTECH HI-Flow 2 | Static Chamber | Dynamic Chamber | GPM | Vent | UAV | OTM-33a | FAST |
|---|---|---|---|---|---|---|---|---|---|
| Hardware Cost | >$50K | ~$40K | >$400 | >$400 | >$5K | ~$50K | >$50K | >$10K | $50K* |
| Range (g/h) | N/A | <1-30,000 | >0.1 | >0.1 | >100 | >100 | >50 | >50 | >1 |
| Uncertainty | N/A | ±10% | -50%, +100% | ±15% | ±40% | N/A | N/A | ±70% | ±50% |
| Size (L) | ~0.3 | ~15 | ~20 | ~20 | ~50 | N/A | ~40 | >1,000 | ~50 |
| Measuring Time (min.) | ~2 | ~3 | >30 | >30 | >10 | >30 | >30 | >10 | ~3 |
| Setup Time (min.) | ~5 | ~5 | >10 | >10 | >10 | >10 | >30 | >30 | ~30 |

**Table 1:** Comparative assessment of commercial (FLIR, SEMTECH Hi-Flow 2, Vent) and research (Chamber, GPM, UAV, OTM-33a) methods used to monitor fugitive methane leaks from orphaned wells. Hardware costs, detection range, accuracy, size, labor and safety are compared for each technology. *The FAST method in this study is currently limited by the high cost of laser trace gas sensors (Picarro, Aeris, etc.) that can be reduced significantly by using cheaper non-laser sensors (i.e. Gas Rover) used in chambers.

General comment 2 Can you please share the caveats that you have added to the manuscript including the line numbers. As you have not included these details in the response I cannot comment if this issue has been addressed.

Yes, here are some comments/caveats we added:

**Abstract L22:** "…low-cost, portable, fast and safe alternative to existing methods with reasonable estimates of orphaned well emissions over a range of leak rates below 40 g/h and within certain geometric and atmospheric constraints."

**L93:** "…provide reasonable estimates of orphaned well emissions under reasonable meteorological conditions."

**L157-162:** "A key assumption of this study is that the effective plume width ($\sigma_0$) derived in Halloran et al. (2014) for smoking oil plumes is applicable to methane dispersion from orphaned wells. While the MDB fan generates turbulent transport similar to Halloran et al., differences in the physical properties of smoking oil and methane—such as buoyancy, diffusion rates, and emission dynamics—could lead to deviations in plume behavior. These potential differences underscore the need for additional experiments designed specifically for methane to validate the use of ($\sigma_0$) under these conditions and further refine the FAST method's applicability."

**L456-462:** "The results for Hooper #41 highlight challenges in measuring methane emissions from variable wells and suggest potential limitations in the FAST method. The variable leak rate led to significant uncertainty in SEMTECH readings (±95.47 g/h), while the FAST method provided more stable estimates (10 ± 10 g/h). However, the fan setup likely failed to fully entrain the emitted gas into the airflow directed toward the sensor, potentially leading to an underestimation of emissions, as supported by SEMTECH data and Figure 10. This limitation is particularly critical for wells with low-height emissions, such as Hooper #41. Future work could address this limitation through controlled release experiments at different heights to optimize the fan and sensor configurations for capturing low-lying plumes."

General comment 3 The addition of section 3.1 is good and clearly explains what was missing in the previous iteration. A necessary addition is the uncertainty of KFAST, this is currently missing and essential to the study. Please add this.

This discussion was added in L347-362 on the recent submission. See below.

"By using the known values of Q from stoichiometry (source rate) and the measured values of C and u during the controlled release experiment, the experimentally determined values of $K_{FAST}$ for different filter angles and fan speeds are estimated via Equation 3. By inverting Equation 3 to solve for $K_{FAST}$, $K_{FAST} = \dfrac{\overline{C_{CL}}\,\overline{u_{CL}}}{Q}$ where the known value of Q and 10 minute averages of $C_{CL}$ and $u_{CL}$ are used to estimate $K_{FAST}$. The resulting values for $K_{FAST}$ are shown as the slopes of the lines in Figure 7 along with the uncertainties resulting from standard error estimates on the linear regression used to generate the line of best fit. As expected, the No Fan scenario has a much higher value of $K_{FAST}$ with higher overall uncertainty due to the variation of the natural wind direction and speed. Without filtering the data by wind direction, the $K_{FAST}$ values are larger (likely due to more dispersion from crosswinds). Furthermore, $K_{FAST}$ values at the low and high fan speeds do not agree, although $K_{FAST}$ is theoretically independent of fan speed (per Equation 4). As more and more crosswind is filtered (Filter Angle approaches 360 degrees), the low and high fan speeds converge to a $K_{FAST}$ of around 0.19 $m^2$, as expected. All fits are done with a 0 intercept and standard errors are used to estimate the uncertainty of $K_{FAST}$. Table 4 shows the resulting experimentally determined values of $K_{FAST}$ and their corresponding uncertainties which were used to estimate emissions and corresponding uncertainties from field measurements."

| Filter Angle | No Fan | Low Fan | High Fan |
|---|---|---|---|
| **0 Degrees** | $1.70 \pm 0.52$ | $0.47 \pm 0.03$ | $0.27 \pm 0.08$ |
| **180 Degrees** | $1.63 \pm 0.51$ | $0.28 \pm 0.01$ | $0.24 \pm 0.03$ |
| **300 Degrees** | $1.11 \pm 0.37$ | $0.20 \pm 0.01$ | $0.19 \pm 0.01$ |

**Table 4:** Values of $K_{FAST}$ in $m^2$ and their associated uncertainties under various filter and fan conditions determined from the Richmond controlled release experiment.

General comment 4 The response is mostly OK apart from the statement "Despite this, the FAST method shows promise for measuring variable emissions more consistently than SEMTECH.". There is no evidence that the emission from Hooper #41 is constant and abandoned wells have shown to have variability on very short timescales. I suggest this sentence is removed.

Thank you, this sentence was removed.

General comment 5 Response is good. Specific comments All ok apart from those listed below.

Thanks.

Original L513-518 – Now P 25 starting "Furthermore, the type…"
This makes a very big assumption that lower cost sensors will have the same accuracy/precision as the Picarro and are sensitive enough to be used to measure ppm-level concentrations. In nearly all cases, this is not true. The NDIR and Gas-Rover will not be able to measure at low concentrations (< 10 ppm) while the Nikira Labs' Portable Methane Gas is still quite expensive (tens of thousands). This whole part of the discussion is highly speculative and should not be included, i.e. from "Furthermore," to "[Portable Methane Analyzer]."

While this is still speculative, it is actually part of undergoing work we are doing now and mentioned as "future work" explicitly. This was added in response to another reviewer's request for more specific technologies, which we have now removed. We have dialed it back to the following:

"Furthermore, the type of methane sensor can be optimized to a more reasonable price point, as $CH_4$ signals near sources are typically high (e.g., > 1 ppm for leaks > 1 g/h). Future work will focus on investigating a wide variety of methane detection technologies to identify more cost-effective and reliable solutions for wide-scale FAST method deployment."

L528
Again this is highly speculative. I would suggest the following statement is removed "However, with the aforementioned simplified setup, the FAST method's setup time could be reduced to match that of the SEMTECH, making it more practical for field deployment."

Removed.

"5. Conclusions"
Several sentences within the "conclusions" section are not backed up by any of the findings in the paper. This section should be comprehensively reviewed. For example, "In the case of the highly variable Hooper #41, where SEMTECH struggled with the well's fluctuating leak rates, the FAST method's larger sampling cross-section and volume resulted in overall lower emissions estimates and relative uncertainty. However, the fan-driven airflow may not fully entrain all emitted gas, particularly from low-height leaks, potentially leading to an underestimation of emission rates for such wells." How do you know the Hi-Flow "struggled"? This sentence should be rewritten. The following sentence stating "Future developments in sensor optimization, including the use of more affordable wind and methane detectors, are expected to further enhance its deployment efficiency and accuracy across diverse field conditions. Further testing is being done to optimize the necessary wind and methane sensors to lower costs and maintain accuracy in order to deploy this technology across the U.S. to quantify fugitive emissions." I understand you are trying to convey that the system could be optimized to overcome some of the current shortcomings but it sounds like an advert for a product and shouldn't be included in a scientific paper as it stands. This should be rewritten.

Noted, this has been updated as follows to be clear, concise and non-advertising:

"In the Texas and Oklahoma field campaigns, the FAST method provided accurate and rapid readings under varying environmental conditions, with errors on the order of 95% of the emission rate across different wind conditions and leak rates. In Texas, where wind speeds were low, only the Low Fan setting was used, and FAST results aligned closely with SEMTECH, within 10%. In Oklahoma, higher wind conditions required both Low and High Fan settings to account for greater natural dispersion. At Hooper #41, where emission rates fluctuated significantly, FAST produced lower overall estimates than SEMTECH, likely due to its larger sampling cross-section averaging out short-term variability. However, fan-driven airflow may not fully entrain all emitted gas, particularly from low-height leaks, which could contribute to an underestimation of emission rates in certain cases.

The FAST method offers a potential alternative to existing technologies such as SEMTECH and FLIR for identifying high-priority orphan wells. Its combination of controlled airflow and real-time methane measurement enables rapid assessments suitable for large-scale monitoring. Ongoing research aims to refine wind and methane sensor integration to improve cost efficiency while maintaining measurement accuracy across diverse field conditions and leak rates."